# Dynamic enhancer landscapes in human craniofacial development

Sudha Sunil Rajderkar [1], Kitt Paraiso[1], Maria Luisa Amaral [2], Michael Kosicki[1], Laura E. Cook [1], Fabrice Darbellay [1,10], Cailyn H. Spurrell [1], Marco Osterwalder [1,11,12], Yiwen Zhu [1], Han Wu [1], Sarah Yasmeen Afzal[1,13], Matthew J. Blow [3], Guy Kelman [1,14], Iros Barozzi [1,15,16], Yoko Fukuda-Yuzawa[1,17], Jennifer A. Akiyama [1], Veena Afzal[1], Stella Tran[1], Ingrid Plajzer-Frick[1], Catherine S. Novak[1], Momoe Kato[1], Riana D. Hunter[1,18], Kianna von Maydell[1], Allen Wang[4], Lin Lin[4], Sebastian Preissl [4,5], Steven Lisgo [6], Bing Ren[7], Diane E. Dickel[1,19], Len A. Pennacchio [1,3,8] & Axel Visel [1,3,9] ✉

The genetic basis of human facial variation and craniofacial birth defects remains poorly understood. Distant-acting transcriptional enhancers control the fine-tuned spatiotemporal expression of genes during critical stages of craniofacial development. However, a lack of accurate maps of the genomic locations and cell type-resolved activities of craniofacial enhancers prevents their systematic exploration in human genetics studies. Here, we combine histone modification, chromatin accessibility, and gene expression profiling of human craniofacial development with single-cell analyses of the developing mouse face to define the regulatory landscape of facial development at tissue- and single cell-resolution. We provide temporal activity profiles for 14,000 human developmental craniofacial enhancers. We find that 56% of human craniofacial enhancers share chromatin accessibility in the mouse and we provide cell population- and embryonic stage-resolved predictions of their in vivo activity. Taken together, our data provide an expansive resource for genetic and developmental studies of human craniofacial development.

The development of the human face is a highly complex morphogenetic process. It requires the precise formation of dozens of intricate structures to enable the full complement of facial functions including food uptake, breathing, speech, major sensory functions including hearing, sight, smell, taste, and nonverbal communication through facial expression. Intriguingly, these functional constraints coincide with substantial inter-individual variation in facial morphology, which humans use as the principal means for recognizing each other. Apart from providing the basis for normal facial variation, early developmental processes underlying facial morphogenesis are highly sensitive to genetic abnormalities as well as environmental effects[1]. Even subtle disturbances during embryogenesis can result in a range of craniofacial defects or dysfunctions[2]. In embryonic facial development, the primary germ layers, as well as the neural crest contribute crucially to the formation of the pharyngeal arches, the frontonasal process and the midface, which in combination give rise to the derived structures of the face[3–6]. The primary palate forms by the fifth week post conception[7] and the development of primary palate derivatives, secondary palate, and many other structures, combined with overall rapid growth, result in a discernable human-like appearance by the tenth week post conception[8]. Genetic or environmental perturbations during these crucial developmental stages are known to result in craniofacial malformations of varying severity and of typically irreversible nature[9–13]. Development of the mammalian face requires a conserved

set of genes and signaling pathways[14], which are regulated by distant-acting transcriptional enhancers that control gene expression in time and space[15–22]. Together with the genes they control, these enhancers are a critical component of mammalian craniofacial morphogenesis. It is estimated that there are hundreds of thousands of enhancers in the human genome for ~20,000 genes[23], and chromatin profiling studies have identified initial sets of enhancers predicted to be active in craniofacial development[15,23,24]. However, these datasets do not cover critical stages of human facial development, such as secondary palate formation. Several single-cell studies have been performed for the developing face in vertebrate and mammalian model systems, as well as some human face tissues[6,25–42]. While these studies cover several specific cell lineages or anatomical subregions of the face, the broad enhancer landscape of mammalian face development at cell-type resolution remains incompletely understood. In part due to the continued incomplete annotation state of the craniofacial enhancer landscape, the number of enhancers that could be mechanistically linked to facial variation or craniofacial birth defects has remained limited[15–21]. With an increasingly refined view of the genetic variation underlying human facial variation[43] and whole genome sequencing as an increasingly common clinical approach for the identification of noncoding mutations in craniofacial birth defect patients[44,45], an expanded and accurate map of human craniofacial enhancers is critical for the interpretation of any noncoding findings emerging from these studies.

Here we provide a comprehensive compilation of regulatory regions from the developing human face during embryonic stages critical for birth defects, including orofacial clefts, along with gene expression and open chromatin signatures at single-cell resolution for the developing mouse face.

## Results

### Epigenomic landscape of the human embryonic face

To map the epigenomic landscape of critical periods of human face development, we focused on Carnegie stages (CS) 18–23, a period coinciding with the formation of important structures, including the maxillary palate, rapid overall growth, and significant changes in the relative proportions of craniofacial structures that impact on ultimate craniofacial shape[8,46,47]. These stages are of direct clinical relevance because common craniofacial defects, including cleft palate and major facial dysmorphologies, result from disruptions within this developmental window (Fig. 1a)[48,49]. To determine the genomic location of enhancers, we generated genome-wide maps of the enhancer-associated histone mark H3K27ac (ChIP-seq), accessible chromatin (ATAC-seq), and gene expression (RNA-seq) from embryonic face tissue for CS18, 19, 22, and 23 (Supplementary Fig. 1, Supplementary Data 1). To extend our compendium to earlier stages, we complemented this data with published H3K27ac peaks (ChIP-seq) from CS13-17 human face tissue and an additional available sample at CS20[24] (Supplementary Data 1; Methods). In total, we observed 13,983 reproducible human candidate enhancers, as defined by the presence of H3K27ac signal in at least two biological samples at any stage between CS13-23 of development (Supplementary Data 2). We examined the correlation between H3K27ac peaks and chromatin accessibility focusing on week 7 (comprising CS18 and CS19), since the largest number of perfectly matched datasets (H3K27ac peaks and chromatin accessibility data from the same biological samples) were available for this stage. We observed that 2225 out of 3182 (70%) of the reproducible H3K27ac peaks overlap at least one ATAC-seq peak derived from the same samples (Supplementary Data 3; Methods).

For an initial assessment of the biological relevance of this genome-wide set of predicted human craniofacial enhancers, we compared it with the large collection of in vivo-validated enhancers available through the VISTA enhancer browser[50]. Among the 130 human craniofacial regulatory elements that have been tested in VISTA

to date and that are annotated for branchial arch, facial mesenchyme, or nose, we identified 38 cases (29%) with overlaps with an enhancer predicted through the present human-derived epigenomic dataset (Supplementary Fig. 2, Supplementary Data 4). A representative example of a validated VISTA craniofacial enhancer is shown in Fig. 1b.

To assess the value of these data for the discovery of additional craniofacial in vivo enhancers in the human genome, we tested 60 candidate human enhancers in a transgenic mouse assay (Supplementary Data 5; Methods). Of these, a total of 28 candidate enhancers were positive for reporter activity, out of which we identified 16 cases of previously unknown enhancers that showed reproducible activity in craniofacial structures. Figure 1c illustrates the rich diversity of craniofacial structures in which these enhancers drive reproducible in vivo activity. Examples include enhancers driving expression in restricted subregions of the medial nasal and mandiblular processes (hs2578), the mandibular (hs2580), the mandibular process and the second pharyngeal arch (hs2724), the maxillary (hs2740), the medial nasal and maxillary processes (hs2741), or the lateral nasal process (hs2752, Fig. 1c). Of the 16 enhancers positive for craniofacial tissues, 8 were simultaneously active in non-craniofacial structures such as the brain or limb, while the remaining 12 out of the total 28 were only positive in non-craniofacial tissues (Supplementary Data 5).

### Developmental dynamics of human craniofacial enhancers

To further assess the biological relevance of the human candidate enhancer sequences identified by our approach, we examined known functions of their presumptive target genes using rGREAT ontology analysis[51]. The identified candidate enhancers are enriched near genes implicated in craniofacial human phenotypes, with 9 of the top 15 terms directly related to craniofacial or eye-associated phenotypes (Fig. 2a, and Supplementary Data 6), including midface retrusion, reduced number of teeth, and abnormality of maxilla.

In a complementary assessment, we explored the putative target genes of the human reproducible enhancers with predictions from publicly available promoter-centric long-range chromatin interaction data for ~19,000 human promoters[52]. This interaction-based mapping strategy identified 3005 chromatin segments containing predicted craniofacial enhancers interacting with the promoters of 2921 nearby genes (Supplementary Data 7; Methods). Across 2263 predicted gene-enhancer pairs with epigenomic enhancer predictions and gene expression data available from identical biological samples, we observed a positive correlation between sample-specific enhancer activity and gene expression levels ($p = 0.00002$; Mann–Whitney $U$ test; see Supplementary Fig. 3, Supplementary Data 8; Methods). We also examined the genome-wide set of human craniofacial candidate enhancers for the presence of noncoding variants implicated in inter-individual variation in facial shape and in craniofacial birth defects through genome-wide association studies (GWAS). We aggregated lead SNPs from 41 studies of normal facial variation and craniofacial disease (Supplementary Data 9; Methods). From 1404 lead SNPs from these studies, we identified 27,386 SNPs in linkage disequilibrium (LD; $r^2 \geq 0.8$) with the lead SNPs for the appropriate populations in the respective craniofacial GWAS. Upon intersection with H3K27ac-bound regions from bulk face tissue between stages CS13-23 (Fig. 1a), we observed a total of 209 predicted enhancer regions overlapping with 605 unique LD SNPs. This LD SNP density represents an enrichment compared to control SNPs not implicated in craniofacial traits (OR = 1.27, $p < 10^{-8}$; Methods). This includes 43 candidate enhancer regions overlapping with 102 unique disease SNPs, and 176 candidate enhancers overlapping with 515 unique SNPs for normal facial variation (Supplementary Data 10).

The activity of individual enhancers can be highly dynamic across developmental stages, supporting that enhancers regulate both spatial and temporal aspects of developmental gene expression[23,53]. To explore the temporal dynamics of human craniofacial enhancers, we

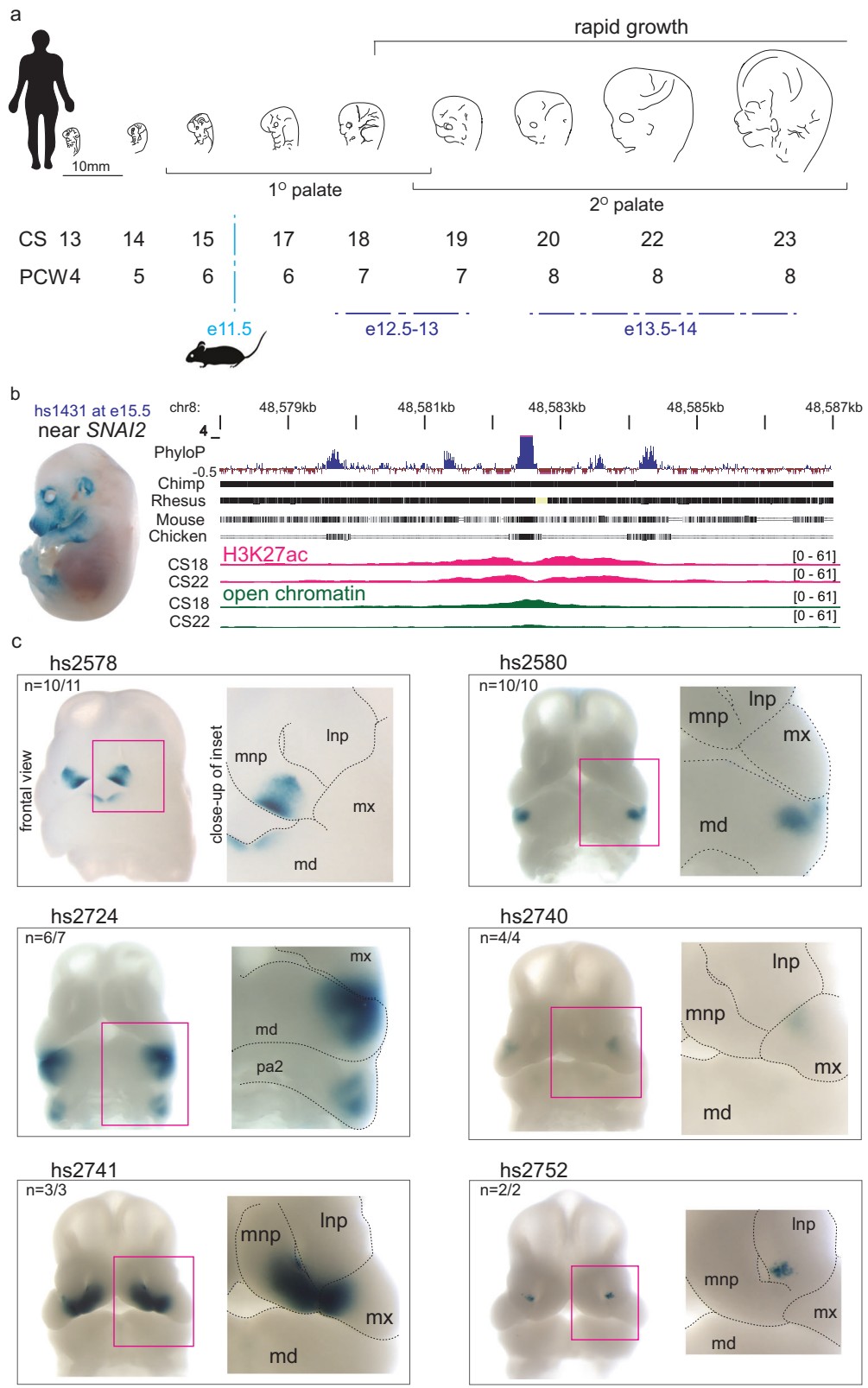

determined the temporal activity profile of all 13,983 human candidate enhancers by week of development, covering gestational weeks 4 to 8 (Fig. 2b; Methods). We found that a small proportion (1624 elements or 11.6%) of elements were predicted to be continuously active (labeled "constant" in Fig. 2b) as enhancers throughout all five weeks. Nearly half (6347) showed narrow predicted activity windows limited to a single week, while another 3,749 showed continuous activity periods covering a subset of the five weeks. A smaller number of enhancers (2236) with predicted non-continuous activities likely contains elements with truly discontinuous activity (e.g., in different subregions of the developing face), and elements not reaching significant signal at some stages, e.g., due to changes in relative abundance of cell types. We note that the analysis of temporal dynamics of subsets of enhancers may potentially be influenced by the variable number of samples

**Fig. 1 | Developmental enhancers in human craniofacial morphogenesis.**
**a** Developmental time points coinciding with critical windows of craniofacial morphogenesis are shown by Carnegie stage (CS) and post-conceptional week (PCW) in humans, and comparable embryonic (**e**) stages for mouse are shown in embryonic days. **b** Representative embryo image at e15.5 for an in vivo-validated enhancer (hs1431) shows positive *lacZ*-reporter activity in craniofacial structures (and limbs). Adjacent graphic shows the genomic context and evolutionary conservation of the region, with H3K27ac-bound and open chromatin regions located

within the hs1431 element. **c** Six examples of human craniofacial enhancers discovered in this study with in vivo activity validated in e11.5 transgenic mouse embryos. Enhancers hs2578, hs2580, hs2724, hs2740, hs2741 and hs2752 show *lacZ*-reporter activity in distinct subregions of the developing mouse face. Lateral nasal process (lnp), medial nasal process (mnp), maxillary process (mx), mandibular process (md), and pharyngeal arch 2 (pa2). *n*, reproducibility of each pattern across embryos resulting from independent transgenic integration events.

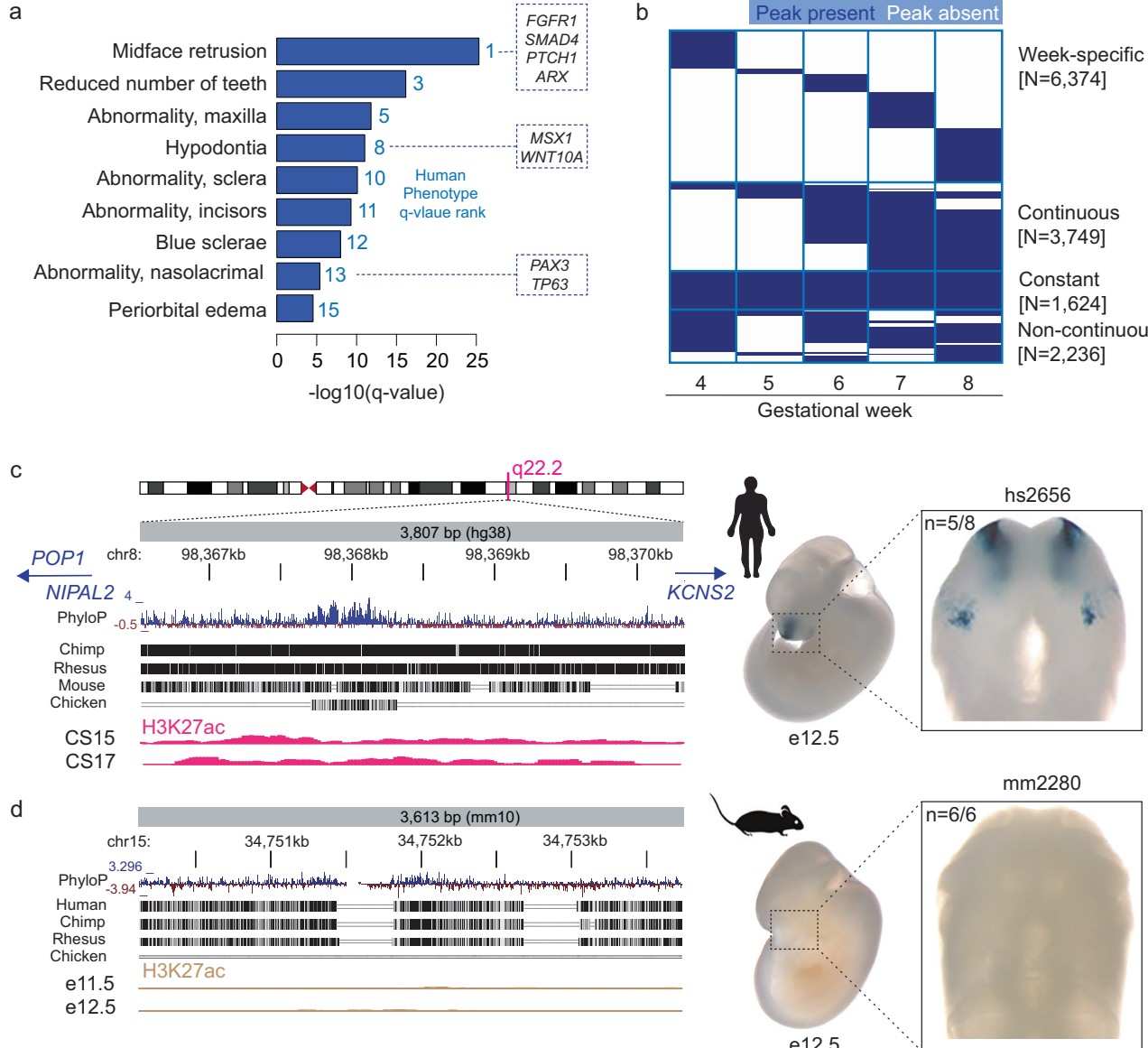

**Fig. 2 | Developmental dynamics and conservation of human craniofacial enhancers.** **a** Results of rGREAT ontology analysis for 13,983 reproducible human craniofacial enhancers, ranked by Human Phenotype *q*-value. The ontology terms indicate that our predictions of human craniofacial enhancers are enriched near presumptive target genes known to play important roles in craniofacial development (examples in boxes). **b** Predicted activity windows of 13,983 candidate human enhancers (rows) arranged by gestational weeks 4–8 of human development (columns). Blue, active enhancer signature; white, no active enhancer signature. Source data are provided as part of Supplementary Data 2 and in the Source Data file.

**c**, **d** Left: genomic position and evolutionary conservation of human candidate enhancer hs2656 (**c**) and its mouse ortholog mm2280 (**d**). The human sequence, but not the orthologous mouse sequence, shows evidence of H3K27ac binding at corresponding stages of craniofacial development (beige tracks). Right: Representative embryo images at e12.5 show that human enhancer hs2656, but not its mouse ortholog mm2280, drives reproducible *lacZ*-reporter expression in the developing nasal and maxillary processes at e12.5. *n*, reproducibility of each pattern across embryos resulting from independent transgenic integration events.

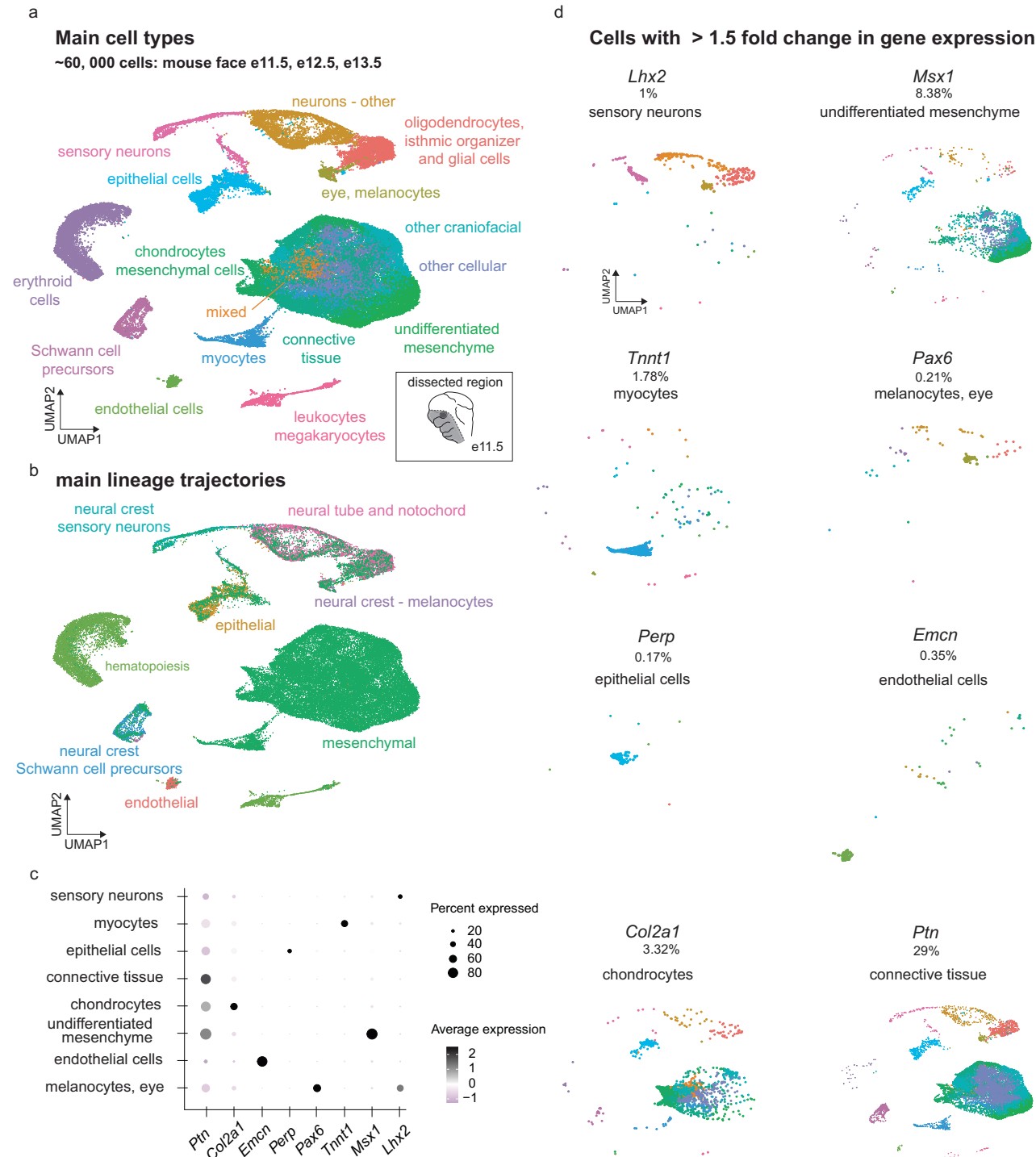

**Fig. 3 | Gene expression in the mammalian craniofacial complex at single-cell resolution. a** Uniform Manifold Approximation and Projection (UMAP) clustering, color-coded by inferred cell types across clusters from aggregated scRNA-seq for the developing mouse face at embryonic days 11.5–13.5, for 57,598 cells across all stages. Cartoon shows the outline of dissected region from the mouse embryonic face at e11.5, corresponding regions were excised at other stages. **b** Same UMAP clustering, color-coded by main cell lineages. **c** Expression of select marker genes in cell types shown **a**. See Supplementary Fig. 11 for additional details. **d** UMAP plots comprising cells with >1.5-fold gene expression for marker genes representing specific cell types as shown in **a** and **c**. Source data are provided as a publicly accessible Seurat/R object file, see Data Availability Statement for details.

or peaks per week. However, we do not observe obvious confounding effects due to these variables within the samples we have analyzed (Supplementary Fig. 4, Supplementary Data 11; Methods). In combination, these datasets provide an extensive catalog mapping the genomic location of human craniofacial enhancers, including their

temporal activity patterns during critical stages of craniofacial development.

To assess the conservation of candidate enhancers identified from human tissues in the mouse model, we compared H3K27ac binding data from human developmental stages CS13-23 to published results

for histone modifications at matched stages of mouse development[23]. The majority (12,179 of 13,983; 87%) of the human candidate enhancers are conserved to the mouse genome at the sequence level, defined by the presence of alignable sequence using LiftOver (UCSC Genome Browser[54]) and that is syntenic relative to surrounding protein-coding genes. Among these conserved sequences, 8257 (59%) of the human candidate enhancers showed H3K27ac binding in the mouse, indicating their functional conservation. The remaining 3922 (28%) regions were sequence-conserved but showed no evidence of enhancer activity in the mouse tissues examined (Supplementary Data 12; Methods), suggesting that these regions are active enhancers in humans only and highlighting the potential value of human tissue-derived epigenomic data for human craniofacial enhancer annotation.

To assess whether the differences in epigenomic signatures between human and mouse translate into species-specific differences in in vivo enhancer activity, we used a transgenic mouse assay to compare the human and mouse orthologs of an enhancer showing an active enhancer signature in the human genome only. We chose a candidate enhancer located near genes *POP1*, *NIPAL2*, and *KCNS2*, located in the *8q22.2* region associated with non-syndromic clefts of the face[55] (Fig. 2c/d). Documented mutations in *POP1* cause Anauxetic Dysplasia with pathognomonic short stature, hypoplastic midface and hypodontia along with mild intellectual disability[56–58]. We generated enhancer-*lacZ*-reporter constructs of the human and mouse orthologs of the candidate enhancer region and used CRISPR-mediated transgene insertion at the H11 safe harbor locus[59,60] to create transgenic mice. Embryos transgenic for the human ortholog (hs2656) show reproducible activity in the developing nasal and maxillary processes at embryonic day (e) 12.5, confirming that the human tissue-derived enhancer signature correctly predicts in vivo activity at the corresponding stage of mouse development (Fig. 2c). In contrast, we did not observe reproducible craniofacial enhancer activity with the mouse orthologous sequence, concordant with the absence of enhancer chromatin marks in mouse at this location (mm2280, Fig. 2d).

## Single-cell transcriptomics of the craniofacial development

To provide a higher-resolution view of the enhancer landscape of craniofacial development, we complemented these detailed maps of human craniofacial enhancers with single-cell-resolved data, with the goal to identify the cell population-resolved activity signatures of individual enhancers. Given the genetic heterogeneity, limited availability, and processing challenges associated with early human prenatal tissues, we performed these studies on mouse tissues isolated from corresponding developmental stages (Fig. 3).

We generated a detailed transcriptome atlas from relevant stages of development and analyzed mouse facial tissue isolated from e11.5, e12.5, and e13.5 by single-cell RNA-seq (see Methods). Applying Uniform Manifold Approximation and Projection (UMAP) non-linear dimensionality reduction for unbiased clustering resulted in 42 primary detectable clusters (Supplementary Figs. 5–8, Supplementary Data 13–14). We analyzed 57,598 cells with a median of 1659 genes expressed per cell. We systematically assigned cell type identities to the resulting clusters (Supplementary Figs. 9–10, Supplementary Data 15–16; Methods) in our final Single-cell annotated Face eXpression dataset (henceforth referred to as *ScanFaceX*), which includes 16 annotated cell types capturing the developing mammalian face and associated tissues (Fig. 3a). Trajectory analyses using Seurat recapitulated the main lineages including epithelial, mesenchymal, endothelial, and neural crest-derived cell types including melanocytes relevant to face development (Fig. 3b). The final annotated cell type clusters showed strong cluster-specific expression of established markers genes relevant to craniofacial development such as *Col2a1* (chondrocytes)[61–63], *Msx1* (undifferentiated mesenchyme)[64–66], *Perp* (epithelial cells)[67,68], *Emcn* (endothelial cells)[69,70], *Lhx2* (sensory neurons)[71,72], *Pax6* (melanocytes)[73,74], *Tnnt1* (myocytes)[75], and *Ptn*

(connective tissue)[76] (Fig. 3c, d, Supplementary Fig. 11). These benchmarking results indicate that *ScanFaceX* provides an accurate single-cell transcriptome reference for relevant stages of craniofacial development that can serve as a foundation for integration with other chromatin data types.

## Differential chromatin accessibility and gene expression

To identify developmental enhancers at single-cell resolution, we performed single-nucleus ATAC-seq (snATAC-seq)[77] on mouse face embryonic tissues at select developmental time points (Fig. 4). Across all stages analyzed, 41,483 cells that passed all quality control steps were considered in the final analysis, and their unbiased clustering resulted in 20 discernable clusters (see Methods). Out of a total of 115,521 open chromatin regions in the snATAC-seq data, we observed 16,564 differential accessible regions (DARs) across 20 separate clusters, indicating that each of the clusters has distinct open chromatin signatures (Supplementary Fig. 12, Supplementary Data 17). Next, we integrated our single-cell open chromatin data with the cell type annotations from *ScanFaceX* single-cell transcriptome data using Seurat-based label transfer (see Methods). Upon integration, a substantial subset of DARs (10,038 out of 16,564; 60%) across 11 annotated clusters for developing craniofacial cell types were retained. Clusters labeled chondrocytes, myocytes and connective tissue, and sensory neurons showed high correlation between the two data types (Fig. 4a, b, Supplementary Figs. 13 and 14; Methods). Chromatin accessibility at putative distal enhancer regions as well as transcription start sites showed distinct cell type specificity. For example, the representative intergenic region near *Isl2* and *Scaper*, and an intronic region of *Lrrk1* differentially active in clusters representing sensory neurons and/or epithelial cells, illustrate the resolution of our data relative to previously available predictions from bulk face tissue[23,78,79] (Fig. 4c). Within the immediate vicinity of these two enhancer regions, we display genes with positive expression in *ScanFaceX* and those that were reported in the OMIM catalog[80,81] as human disease-causing. Both *Isl2* and *Aldh1a3* are highly expressed in sensory neurons and epithelial cell clusters, respectively, in *ScanFaceX* data (Fig. 4c). *Isl2* has been shown to be selectively expressed in a subset of retinal ganglion cell axons that have important functions in binocular vision[82]. Allelic variants and mutations in *SCAPER* cause intellectual disability with retinitis pigmentosa in humans[83–85]. The *Lrrk1* intronic element is near *Aldh1a3*, a gene adjacent to *Lrrk1;* mutations in the orthologous *human ALDH1A3* cause an autosomal recessive form of isolated microphthalmia[86–89]. These putative enhancer regions near *Isl2* and *Scaper*, and in the intron of *Lrrk1* drive reproducible *lacZ*-reporter activity in the developing mouse face at e11.5 in anatomical regions where neuronal and epithelial cell types are expected to be found (mm2285 and mm2282, Fig. 4c). Notably, the spatial expression pattern of mm2285 and mm2282 is consistent with the expression of *Isl2* in cranial ganglia[90,91], and the expression of *Aldh1a3* in the retina and the nasal epithelium[92] in similar developmental windows in mice in vivo. In an additional example, an enhancer near the promoter region of *Mymx*, which is exclusively active in the myocyte cluster, coincides with *Mymx* expression in myocytes in *ScanFaceX* (Supplementary Fig. 15).

To facilitate utilization of the full set of genome-wide, cell type-resolved enhancer predictions, we used these mouse tissue-derived single-cell enhancer predictions in combination with our human bulk tissue-derived enhancer catalog, to generate a Single-cell annotated Face eNhancer (*ScanFaceN*) catalog of human enhancer regions with predicted activity profiles across craniofacial cell types (Supplementary Data 18–20). The majority (7899 of 13,983; 56%) of human tissue-derived facial candidate enhancers overlap with an accessible chromatin region in at least one cluster of our *ScanFaceN* catalog, and 2339 (30%) of these regions overlap with DARs in *ScanFaceN*.

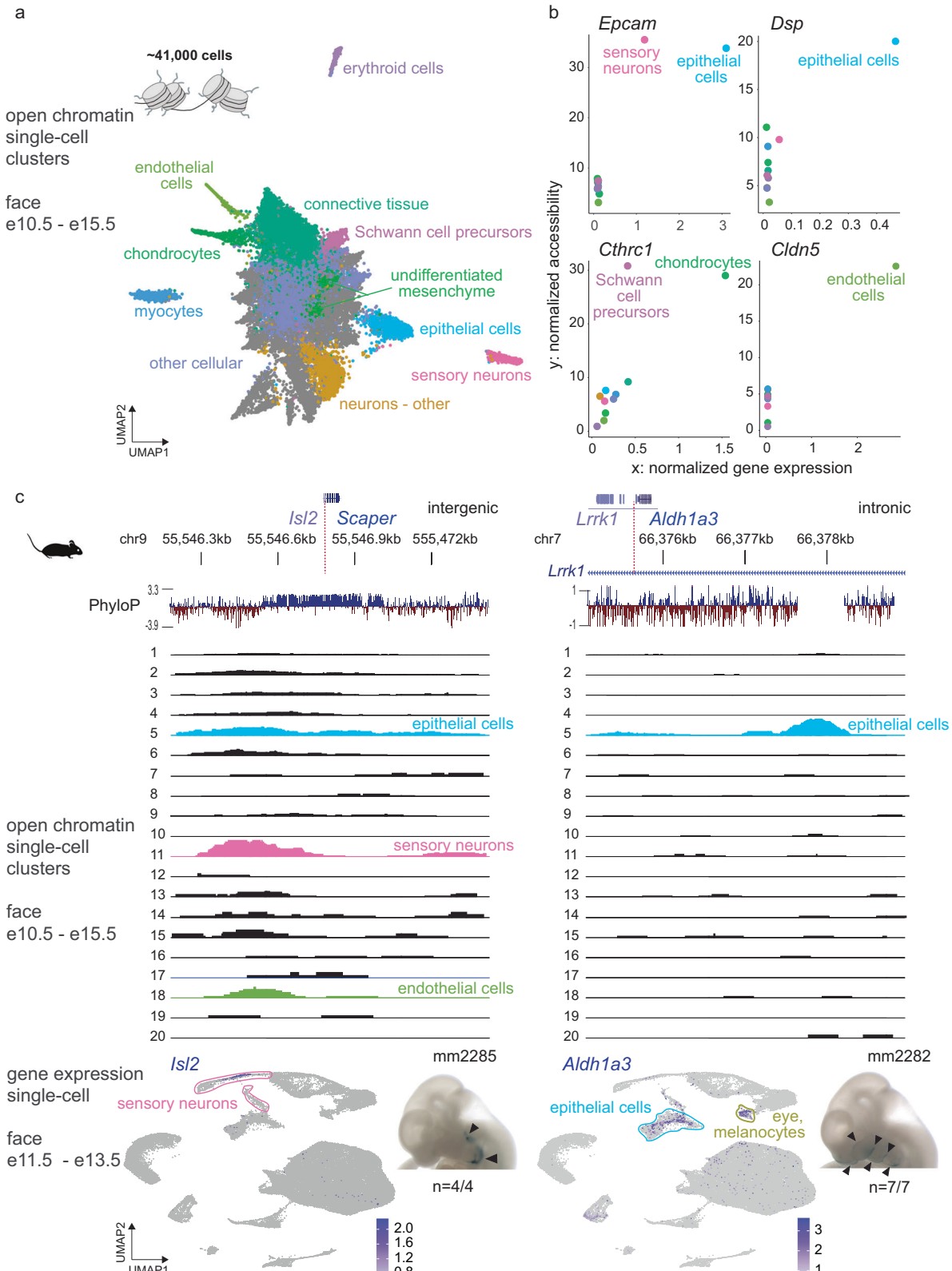

## Cell population-resolved enhancer activity predictions

To explore the relationship between predicted cell type specificities of enhancers and their respective spatial in vivo activity pattern during craniofacial development, we intersected the *ScanFaceN* DARs from the 11 main *ScanFaceX*-matched clusters with craniofacial enhancers validated in vivo and curated in the VISTA Enhancer Browser[50] (Fig. 5a). We observed general correlations between cluster-specific accessibility and spatial in vivo patterns among 77 formerly validated VISTA enhancers that showed chromatin accessibility in at least one of the 11 main clusters. For example, the predicted connective tissue-mesenchymal cluster (cluster 2) of the craniofacial snATAC-seq tends to group VISTA enhancers with activity specific to the branchial arches (Fig. 5b). Despite broad correlations, we observed considerable heterogeneity of spatial patterns within most clusters. For example, the

**Fig. 4 | Differential chromatin accessibility at craniofacial in vivo enhancers correlates with expression of nearby genes. a** Unbiased clustering (UMAP) of open chromatin regions from snATAC-seq of the developing mouse face for stages e10.5–15.5 for ~41,000 cells. The cell types are assigned based on label transfer (Seurat) from cell type annotations of the *ScanFaceX* data. **b** Correlation between normalized gene expression (*x* axis) from *ScanFaceX* and normalized accessibility (*y* axis) from snATAC-seq for select genes (*Epcam, Dsp, Cthrc1, Cldn5*) and their transcription start sites with the highest correlation evident in relevant cell types. **c** Genomic context and evolutionary conservation (in placentals) for corresponding regulatory regions in the vicinity of the *Isl2/Scaper* locus, and an intronic distal

enhancer within *Lrrk1*. Tracks for individual snATAC-seq clusters from developing mouse face tissue (e10.5 to e15.5), with cluster-specific open chromatin signatures for relevant annotated cell types are shown for the same genomic regions. Colors in **b** and the individual snATAC-seq tracks in **c** correspond to the color code used in **a**. UMAP of *ScanFaceX* data shows expression of *Isl2* and *Aldh1a3* (gene adjacent to *Lrrk1*) in expected cell types. Images for a representative mouse embryo at e11.5 for both loci show validated in vivo *lac-Z*-reporter activity of the respective regions; black arrowheads point towards stained regions. *n*, reproducibility of each pattern across embryos resulting from independent transgenic integration events. Source data for 4b are provided as a Source Data file.

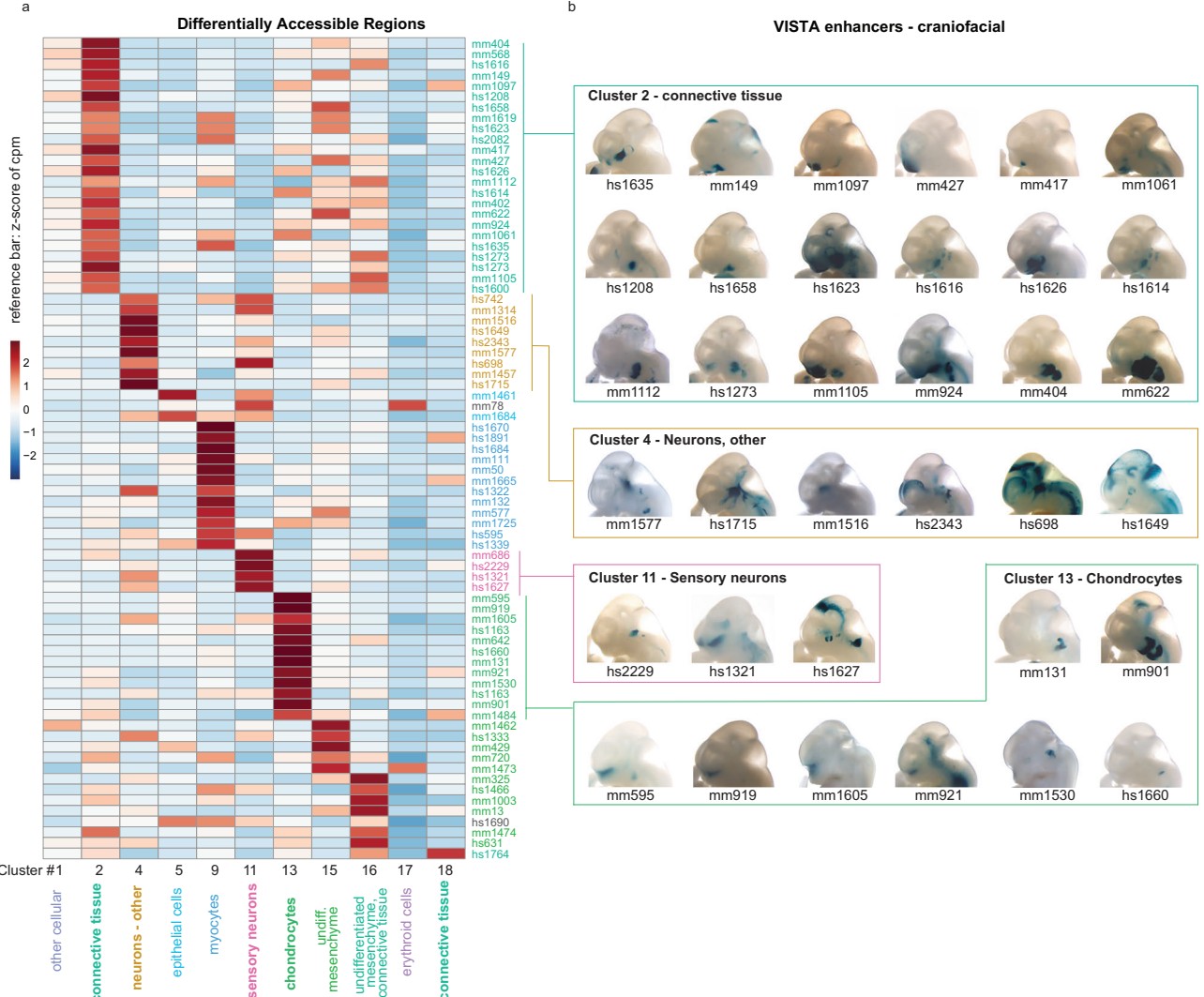

**Fig. 5 | Correlating cell population-resolved enhancer signatures with enhancer in vivo activity patterns. a** Heatmap indicates the chromatin accessibility of 77 craniofacial in vivo VISTA enhancers in 11 major clusters representing predicted cell types. cpm: counts per million. **b** Representative images of transgenic embryos

from VISTA Enhancer Browser, showing in vivo activity pattern of 35 selected enhancers at e11.5. Embryo images are grouped by example cluster types from **a** in this retrospective assignment. Source data for 5a are provided as a Source Data file.

chondrocyte cluster (cluster 13) has multiple VISTA enhancers with activity in the midface, paranasal regions, and/or a region at the junction of the developing forebrain and nasal prominences that may constitute the developing cartilaginous regions of the face (Fig. 5b). These observations underscore the spatiotemporal complexity of craniofacial morphogenesis, which relies on intricate cellular processes in combination with highly regionalized regulatory cues.

**Craniofacial enhancer activity at single-cell resolution**

To explore whether craniofacial enhancer activity can be quantitatively assigned to specific cell types in vivo, we generated transgenic mice in which selected craniofacial enhancers were coupled to a fluorescent *mCherry* reporter gene (Fig. 6a). We examined three different craniofacial enhancers (hs1431, hs746 and hs521), two of which (hs1431 and hs746) we formerly demonstrated to be required for normal facial

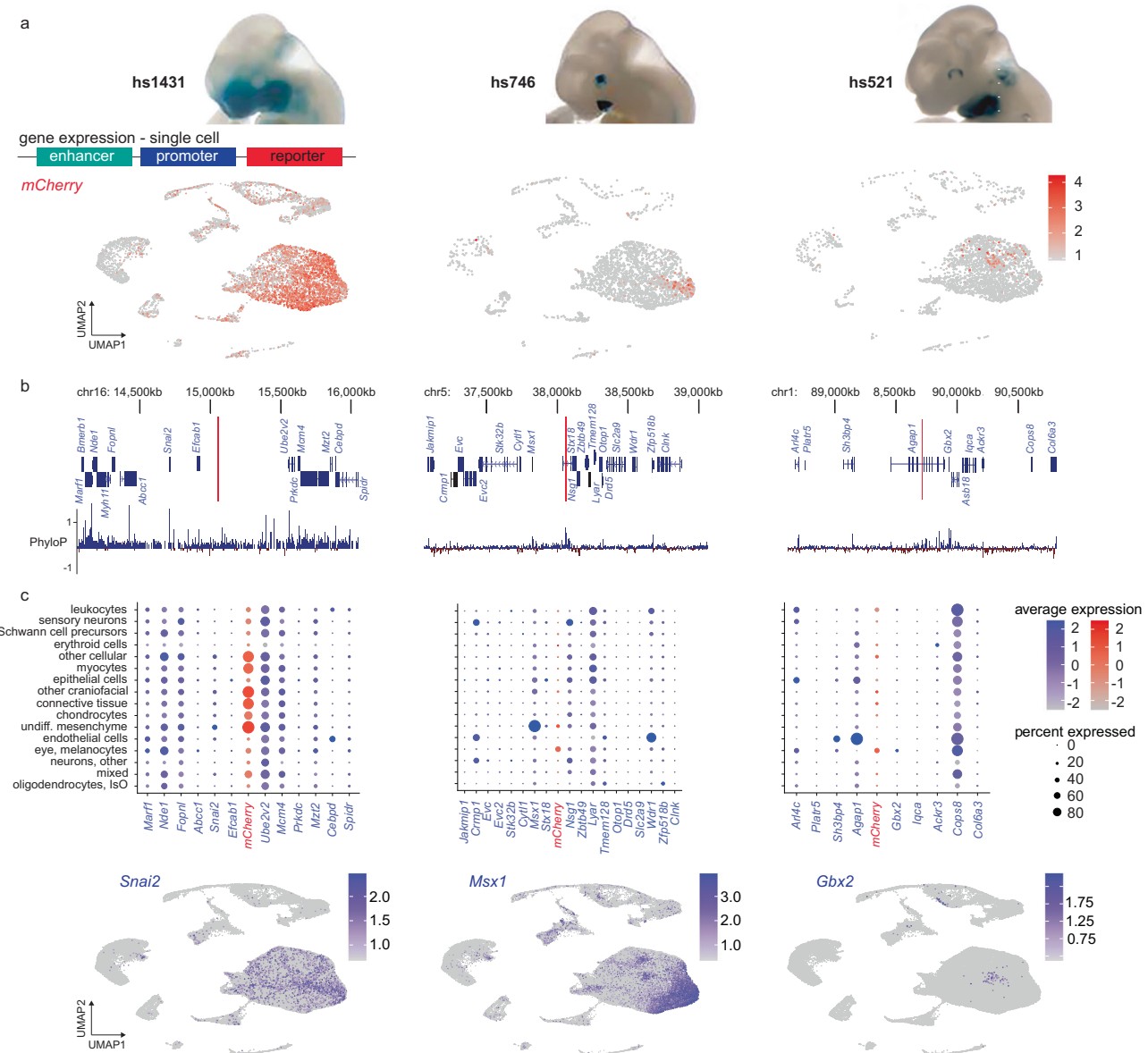

**Fig. 6 | Enhancer activity at single-cell resolution. a** in vivo activity pattern of select craniofacial enhancers (hs1431, hs746, hs521) at e11.5, visualized by *lacZ*-reporter assays (top). In separate experiments, the same enhancers were coupled to a *mCherry*-fluorescent reporter gene and examined by scRNA-seq of craniofacial tissues of resulting embryos. UMAPs show enhancer-driven *mCherry* expression (see Fig. 3a for reference). **b** Location of enhancers hs1431, hs746 and hs521 in their respective genomic context (red vertical lines), along with protein-coding genes within the genomic regions and local conservation profile (PhyloP). **c** Seurat-based average expression of genes in the vicinity of the respective enhancers, and pro-portion (percent) of cells expressing those genes in annotated cell types. Enhancer-driven *mCherry* signal is plotted in the center in between the names of the two genes whose promoters are closest to its location within the genome. For example,

for hs1431, mCherry is highly expressed (indicated by red color intensity) in clusters labeled "other cellular", "myocytes", "skeletal, other", "connective tissue", and "undifferentiated mesenchyme", while it is also expressed in a larger proportion of cells (indicated by greater diameter of the circles) in those same clusters. In the same plot, *Snai2* is highly expressed (indicated by blue color intensity) in a subset of cells (indicated by lesser diameter of circles) in identical clusters as compared to mCherry. Bottom panels show the expression of *Snai2*, *Msx1*, and *Gbx2* as likely candidate target genes for each of the enhancers hs1431, hs746 and hs521 across UMAPs. undiff. undifferentiated, IsO: Isthmic Organizer Cells. Source data are provided as a publicly accessible Seurat/R object file, see Data Availability State-ment for details.

development[15] (Fig. 6b). In all cases, we isolated craniofacial tissue from transgenic reporter embryos at e11.5 and performed scRNA-seq (Fig. 6a). For hs1431, near *Snai2*, which is active across many regions of the developing face, *mCherry* expression is observed across almost all cell clusters, indicating that hs1431 is broadly active across multiple cell types during craniofacial development (Fig. 6c). In contrast, hs746 which is in the vicinity of *Msx1*, is primarily active in a cluster predicted to represent undifferentiated mesenchyme and in a subset of cells expressing *Msx1* in *ScanFaceX*, a gene shown to regulate the

osteogenic lineage[93]. Similarly, based on *ScanFaceX* annotations, enhancer hs521, located near *Gbx2*, is primarily active in a subset of predicted mesenchymal cells and chondrocytes, and its activity coin-cides with a subset of cells expressing *Gbx2* (Fig. 6c), a gene known to be active in the developing mandibular arches[6]. Together, these data illustrate how purpose-engineered enhancer-reporter mice can be used to validate and further explore the in vivo activity patterns of craniofacial enhancers identified through genome-wide single-cell profiling studies.

## Discussion

The lack of data from primary tissues and incomplete mapping of human developmental enhancers in craniofacial morphogenesis has been a challenge in the systematic assessment of the role of enhancers in craniofacial development and disease. In the present study, we have generated human bulk and mouse single-cell data to create a comprehensive compendium of enhancers in human and mouse development, including temporal profiles and predictions of cell type specificity. We identify major cell populations of the developing mammalian face, along with corresponding genome-wide enhancer profiles. While many predicted enhancers show conserved epigenomic signatures indicating an active enhancer state in both mouse and human, we also observed elements with human-specific enhancer activity signatures, suggesting that the human but not the mouse ortholog is an active in vivo enhancer. We also provide additional predictions of regions with human-specific enhancer signatures that show no functional conservation in mice that can be identified by profiling human tissues. We observed that enhancer hs2656, but not its mouse ortholog mm2280, shows craniofacial in vivo activity in transgenic mice. This is consistent with an epigenomic enhancer signature at this element in human, but not in mouse tissue. These lineage-specific differences in epigenomic signature and in vivo activity are likely due to sequence differences within the enhancer element itself, which may affect transcription factor binding sites or other functionally critical motifs embedded in the enhancer. For example, within the most conserved 425 bp core sequence of enhancer hs2256, 31% of the nucleotide positions show differences between humans and mouse, which include binding sites for transcription factors that are important for craniofacial development, such as TFAP2B and TCF4[94–96]. While human-specific signatures would need to be validated in suitable human tissue- or cell-based assays to conclusively confirm bona fide lineage-specific in vivo activity, these data suggest that profiling human tissues is an effective way to identify candidate regions with human-biased enhancer signatures. Our compendium of human craniofacial enhancers expands previously reported[24,33] human craniofacial enhancer catalogs, by ~5000 newly identified enhancers for weeks 7–8 of human craniofacial development primarily identified in this study. When comparing with craniofacial enhancers identified in previous studies, we find that our data provides independent confirmation for 37% of reported primate enhancers and 15% of human-biased enhancers[16]. Of the 13,983 reproducible human enhancers described in this study, 47% showed evidence of enhancer-associated RNA signatures in the FANTOM5 database[97,98]. In contrast, when restricting this analysis to a more differentiated craniofacial cell type available in FANTOM5 (human embryonic palatal mesenchyme)[97–99], we observed enhancer RNA signatures for only 3.8% of our 13,983 predicted enhancers, likely reflecting that this cell type is only one of many that were present in our tissue samples (Supplementary Data 2 and 21). Generally, the imperfect overlap of craniofacial enhancers identified in some of these studies may be due to differences in epigenomic profiles from primary tissues comprising the entire face versus in vitro differentiation of a specific lineage such as neural crest or palatal mesenchyme. Additional possible sources of variation include differences in experimental modalities (H3K27ac binding versus measurements of enhancer RNA), and imperfect matching of in vivo developmental stages with in vitro models. In this study, we leveraged genome-wide profiling of H3K27ac binding for the identification of enhancers. The tissue-specific validation rate we observe is comparable to that we observed in other studies using similar methods for the prediction of in vivo enhancer activities[23]. We note that alternative experimental approaches that measure noncoding RNAs or massively parallel reporter assays with or without mutational screens can also be used for identifying putative enhancer elements and may be useful for capturing additional craniofacial candidate enhancers[100,101].

Our data illustrate the considerable temporal dynamics of human craniofacial enhancers, a critical aspect for understanding the developmental timing of enhancer activity related to specific phenotypes such as clefts and mid-facial deformities. As clinical sequencing becomes increasingly common and accessible to both patients and the medical community, our data may serve as an essential resource to address the gaps in understanding the potential pathogenicity of regulatory variants.

The single-cell resources generated through this study, *ScanFaceX* for gene expression and *ScanFaceN* for enhancers, contain a total of 115,521 candidate enhancers as defined by chromatin accessibility, including 10,038 that show differential chromatin accessibility for major cell types in face morphogenesis. While previous single-cell studies of the developing face from other animal models have described extensive annotations for ectomesenchyme, we find that the complexity of cell types in the developing mouse face poses some challenges in this respect. In particular, in comparing several mouse orthologs of the embryonic zebrafish ectomesenchymal markers[31] expressed in *ScanFaceX* that show relatively high accessibility in *ScanFaceN* in neural crest-derived populations (Supplementary Fig. 16), regional identities marked by specific genes are not obviously delineated in *ScanFaceX*. These differences in cell type distributions and marker gene activity may be explained by the extent of differentiation, growth rate, evolving cell states, and developmental timing underlying craniofacial morphogenesis. One of the limitations of present methods is the ability to capture low-expressing genes or rarer cell populations among other technical and statistical challenges[102,103]. We also note that utilizing cell type annotations from *ScanFaceX* and integrating those with single-cell open chromatin data provides correlative but not definitive evidence for the target genes of a given enhancer, which requires verification through complementary experimental methods[104–106]. We demonstrated how engineered mice can be used to study these enhancers in vivo at single-cell resolution. Using a transgenic reporter assay coupled to single-cell RNA-seq, we defined the activity of three craniofacial enhancers during embryonic development at single-cell resolution. This approach illustrates how these methods can be combined to determine the in vivo specificity of individual enhancers and relate their activity to cell type-specific expression of their putative target genes. We note that in vivo transgenic reporter assays can demonstrate that an enhancer is sufficient to drive expression in a tissue or cell type of interest, but integration into a safe harbor locus such as H11 removes the enhancer from the full epigenomic and three-dimensional context of its native locus[107]. Therefore, reporter expression may not fully recapitulate the full endogenous activity of a given enhancer in its original genomic location.

All of these data are also available in FaceBase and the VISTA Enhancer Browser for community use[15,79,108]. In summary, our work provides a multifaceted and expansive resource for studies of craniofacial enhancers in human development and disease.

## Methods

### Ethics statement

This research complies with all relevant ethical regulations. All aspects involving human tissue samples were reviewed and approved by the Human Subjects Committee at Lawrence Berkeley National Laboratory (LBNL) Protocol Nos. 00023126 and 00022756. All animal work was reviewed and approved by the LBNL Animal Welfare and Research Committee.

Human embryonic face samples were obtained from the Human Developmental Biology Resource's Newcastle site (HDBR, hdbr.org), in compliance with applicable state and federal laws. The National Research Ethics Service reviewed the HDBR study under REC Ref 23/NE/0135, and IRAS project ID: 330783 in compliance with requirements

from the National Health Services for research within the UK and overseas. HDBR is a non-commercial entity funded by the Wellcome Trust and Medical Research Council. Fetal tissue donation is confidential, anonymized, completely voluntary with fully informed and explicitly documented written consent, and the participants do not receive compensation. In accordance, no identifying information for human samples in this study was shared by HDBR. More information about HDBR policies and ethical approvals can be accessed at https://www.hdbr.org/ethical-approvals.

## Human samples
Primary data from embryonic whole face samples at post conception weeks 7 and 8 were generated in this study. Whole face region, excluding eyes was dissected at HDBR (Supplementary Fig. 1), and all embryonic samples were shipped on dry ice and stored at −80 °C until processed. Embryos of both sexes were included in the experiments. However, we did not consider embryo sex as a variable in our studies since craniofacial development is expected to show minimal differences at these early stages of development. ChIP-seq data for three samples at Carnegie stage (CS)18, one sample at CS19, two samples at CS22 and one sample at CS23 are presented in this study, along with accompanying ATAC-seq data for two samples at CS18, one sample at CS19, one sample each at CS22 and CS23. RNA-seq data for four samples at CS18, one sample at CS19, seven samples at CS22, and four samples at CS23 were generated in this study and analysis from a subset of these is presented. Processed data for CS13-17, and CS20 was obtained from previously published studies[24] and included in our downstream integrative analyses. All datasets are listed in Supplementary Data 1.

## Animal studies and experimental design
Mice used for this study were housed at the LBNL Animal Care Facility, which is fully accredited by AAALAC International. Mice were housed on a 12-h light-dark cycle in standard micro-isolator cages on hardwood bedding with enrichment consisting of crinkle-cut naturalistic paper strands. Mice were maintained on ad libitum PicoLab Rodent Diet 20 (5053) and water supply with 30–70% environmental humidity and temperature of 20–26.2 °C. All mice were health checked and monitored daily for food and water intake by trained personnel. Animals of both sexes were used in the analysis. Sample size selection strategies and scoring criteria were followed based on our experience of performing transgenic mouse assays for ~3000 published enhancer candidates[59,60].

## Transgenic mouse assays in vivo
60 candidate human enhancer elements were selected based on a combination of criteria, including overlap with ATAC-seq peaks, strength of H3K27ac active enhancer signatures, non-mouse annotated regions, and vicinity of genes with known or proposed roles in craniofacial development based on human genetics and/or mouse knockout studies (e.g., genes listed under term "abnormality of the face"; HP:0000271 in Human Phenotype Ontology[109] or "craniofacial abnormalities"; MP:0000428 in the Mammalian Phenotype Browser[110]). Mouse enhancer elements mm2280, mm2281 mm2282, and mm2285 were selected based on conservation criteria or predicted from single-cell gene expression readouts and single-cell chromatin accessibility profiles. Transgenic enhancer-reporter assays were performed per established protocols[59,60]. Briefly, a minimal *Shh* promoter and reporter gene were integrated into a non-endogenous, safe harbor locus[60] in a site-directed transgenic mouse assay. The selected genomic region corresponding to the selected enhancer element was PCR-amplified from human or mouse genomic DNA where applicable; the PCR amplicon was cloned into a *lacZ*-reporter vector (Addgene #139098) using Gibson assembly (New England Biolabs)[111]. The final transgenic vector consists of the predicted

enhancer–promoter–reporter sequence flanked by homology arms intended for the *H11* locus in the mouse genome. Sequence of the cloned constructs was confirmed with Sanger sequencing or MiSeq. Transgenic mice were generated using our pronuclear injection protocol[60]. Briefly, sgRNAs (50 ng/µl) targeting the *H11* locus and Cas9 protein (Integrated DNA Technologies catalog no. 1081058; at final concentration of 20 ng/µl) was mixed in microinjection buffer (10 mM Tris, pH 7.5; 0.1 mM EDTA). The mix was injected into the pronuclei of single-cell stage fertilized FVB/NJ (Jackson Laboratory; Strain#:001800) embryos obtained from the oviducts of super-ovulated 7–8 weeks old FVB/NJ females mated to 7–8 weeks old FVB/NJ males. The injected embryos were cultured in M16 medium supplemented with amino acids at 37 °C under 5% $CO_2$ for ~2 h and transferred into the uteri of pseudo-pregnant CD-1 (Charles River Laboratories; Strain Code: 022) surrogate mothers. Embryos were collected for downstream experiments at embryonic days 10.5 through 15.5 (Theiler stages 17–23). Beta-galactosidase staining was performed in our standardized pipeline with the following modification. Embryos were fixed with 4% paraformaldehyde (PFA) for 30 min for stages e11.5 and 12.5 while rolling at room temperature. The embryos were genotyped for the presence of the transgenic construct. Embryos positive for transgene integration into the *H11* locus and at the correct developmental stage were considered for comparative reporter gene activity across respective stages and were imaged on a Leica MZ16 microscope. Genomic coordinates for VISTA enhancer hs2656 (Fig. 2); enhancer mm2280 (Fig. 2), mm2282 and mm2285 (Fig. 4), and mm2281 (Supplementary Fig. 15) are shown in Supplementary Data 5 and 22, respectively.

For transgenic experiments demonstrating enhancer activity at single-cell resolution and involving hs1431, hs746 and hs521 (Fig. 6), a combination of *Hsp68* promoter and *mCherry* reporter were used.

## ChIP-seq
Chromatin immuno-precipitations were performed using established methods in our laboratory[112]. Briefly, frozen and non-crosslinked face tissue was dissociated in PBS by pipetting until homogenized and crosslinked with 1% formaldehyde at room temperature. Cells were lyzed, and chromatin was sonicated using a Bioruptor device (Diagenode) to obtain fragments with an average size ranging between 100 and 600 bp. Input sample was set aside and stored appropriately, Protein A and G Dynabeads (Invitrogen) were added to the sample, and chromatin was incubated for 2 h at 4 °C with 5 µg of anti-H3K27ac antibody (Active Motif, Cat# 39133, Lot 01613007). Immunocomplexes were sequentially washed, and the immunoprecipitated DNA complexes were eluted in an SDS buffer at 37 °C for one hour. Samples were reverse-crosslinked with Proteinase K overnight at 37 °C. DNA was purified with a ChIP DNA clean concentrator (D5205 Zymo Research), and a KAPA SYBR Green qPCR mix was used to assess the presence of H3K27 acetylated regions versus negative control regions. DNA was quantified using Qubit, and size distribution and DNA concentration of the samples were assessed on the Agilent Bioanalyzer. Illumina TruSeq library preparation kit was used for downstream library preparation, and libraries were sequenced as single-end 50 bp reads on an Illumina HiSeq2500. ChIP-seq data was analyzed using the ENCODE histone ChIP-seq Unary Control Unreplicated pipeline (https://www.encodeproject.org/pipelines/ENCPL841HGV/) implemented at DNAnexus (https://www.dnanexus.com). Briefly, reads were mapped to the human reference genome version hg38 using BWA (v0.7.7) and sorted bam file generated using samtools (v0.1.19). For the ChIP-seq datasets at CS13-15, CS17 and CS20[24], publicly available and post-mapped TagAlign files were used. Peak calling was performed using MACS2 (v2.2.4; --broad flag, q-value < 0.05); upon broad peak calling and applying the FDR filter, bed files were combined and merged using bedtools[113]. A combined peak set was called by merging peaks from all samples, and overlapping peaks for each sample were

counted using overlap_peaks.py. Merged peaks within 1 kb of transcription start sites as defined by GENCODE were removed, resulting in 70,075 distal peaks. Of those, 13,983 peaks were present in at least two samples in each embryonic week which were retained for final analysis. For a breakdown of samples as well as peaks per week, see Supplementary Data 2, 8, and 11.

We note that the use of human embryonic tissue samples, which are typically derived from individual or a small number of fetal tissue donations, can introduce variability regarding tissue dissection and genetic heterogeneity. While some of these sources of variation are unavoidable, we tried to minimize potential batch effects. To make the analysis as comparable as possible, we downsampled the number of input reads and the read length to a common denominator (15 million and 50 bp, respectively), and used the standard ENCODE peak-calling pipeline. To assess the possible presence of batch effects between data from these studies, we compared temporal transitions between weeks (Supplementary Fig. 4). In this analysis, we did not observe discontinuities specifically associated with the transition time points between batches. While we cannot exclude the presence of some batch effects, this result suggests that study-specific batch effects do not confound our temporal dynamics analysis in major ways.

## ATAC-seq

Embryonic samples were processed for ATAC-seq using standard methods[112]. In short, harvested tissues were lysed, centrifuged for 10 mins at $500 \times g$, at 4 °C, and the resulting cell pellet was treated with the Nextera DNA transposase Tagment DNA Enzyme (Catalog number: 20018705), and the transposed DNA was eluted using Qiagen MinElute PCR purification kit. Samples were then PCR-amplified using the NEB Next High-Fidelity 2xPCR Master Mix (catalog number: NEBE6040SEA) with Nextera PCR primers 1 (AATGATACGGCGACCACCGAGATCTAC ACNNNNNNNNNTCGTCGGCAGCGTC) and 2 (CAAGCAGAAGACGGCAT ACGAGATNNNNNNNNGTCTCGTGGGCTCGG) and DNA was purified as described above. The eluted library was analyzed for quality in a Bioanalyzer High Sensitivity assay, and samples were subsequently deep sequenced on an Illumina HiSeq2500. ATAC-seq data was analyzed using the ENCODE ATAC-seq (unreplicated) pipeline (https://www.encodeproject.org/pipelines/ENCPL344QWT/). Briefly, reads were aligned with the Bowtie2 aligner and filtered to remove unmapped and non-primary alignments, low quality reads as well as PCR duplicates. A subsample of 15 million reads was used as input to peak-calling, adjusted for Tn5 shift reads and sets of biological samples were assembled along with pseudoreplicates. Peak calls excluded ENCODE blacklist regions[114] and peaks were assessed at an Irreproducible Discovery Rate of 0.05.

## RNA-seq

Samples were processed for RNA-seq, and libraries were generated with established protocols[112,115]. Briefly, RNA was isolated from the dissociated face tissue using TRIzol Reagent (Life Technologies), all samples were DNase-treated (TURBO DNA-free Kit, Life Technologies), and assessed for quality (RNA 6000 Nano Kit, Agilent) on a 2100 Agilent Bioanalyzer. TruSeq Stranded Total RNA with Ribo-Zero Human/Mouse/Rat kit (Illumina) was used to prepare RNA-seq libraries according to manufacturer's protocol. RNA-seq libraries were depleted of high molecular weight products in an Illumina Resuspension Buffer and by incubating in 60 μL Agencourt AMPure XP beads for 4 min. AMPure beads were pelleted, and washed twice with 80% ethanol, and the DNA was eluted per manufacturer's instructions. RNA concentration and quality of the RNA-seq libraries were assessed using a 2100 Bioanalyzer with the High Sensitivity DNA Kit (Agilent), and libraries were sequenced as single-end 50 bp reads on an Illumina HiSeq2500.

RNA-seq data was analyzed using the ENCODE RNA-Seq (Long) Pipeline-1 replicate pipeline (https://www.encodeproject.org/pipelines/ ENCPL002LSE/) implemented at DNAnexus (https://dnanexus.com). Briefly, reads were mapped to the reference genome using STAR align (V2.12). Genome-wide coverage plots were generated using bam to signals (v2.2.1). Gene expression counts were generated using RSEM (v1.4.1). Human datasets were analyzed using human reference genome version hg38, and GENCODE v24 gene annotations. Mouse datasets were analyzed using mouse reference genome version mm10 and GENCODE M4 gene annotations.

## rGREAT ontology analyses

To identify human phenotype ontology terms enriched in our list of 13,983 reproducible human craniofacial enhancers, we ran rGREAT[51] (Bioconductor version: Release 3.17) that performs GREAT[116] analysis (http://great.stanford.edu) on noncoding regions to predict their functions based on annotations of nearby genes. The following parameters were used from the GREAT tool: a default of 5 kb upstream and 1 kb downstream basal plus extension for proximal regulatory regions, up to 10 kb for distal regions, and curated regulatory domains were included. A background of whole genome hg38, a cut-off based on binomial false discovery rate <0.01, and fold enrichment >2 was applied to retain the top terms (Supplementary Data 6).

## Enhancer-target gene predictions

We intersected our list of 13,983 reproducible human enhancers with publicly available long-range chromatin interaction data derived from promoter capture HiC for -19,000 promoters in human embryonic stem cells[52]. Genomic coordinates of the interacting fragments were converted to hg38, the predicted target gene and extent of overlap with the human enhancers from this study are reported in Supplementary Data 7. For 3005 chromatin segments containing predicted human craniofacial enhancers, and interacting with the promoters of 2921 genes, we performed Spearman's Ranked Correlation Coefficient (SRCC) analysis between enhancer signal intensities (H3K27ac ChIP-seq, Trimmed Mean of M-values normalized) and gene expression counts (RNA-seq) of the assigned target genes (Supplementary Data 7) for predicted enhancer:target gene pairs versus all other pairs. We performed this analysis for combined as well as individual activity windows shown in Fig. 2b for a subset of matched samples, i.e., five instances where enhancer predictions and gene expression data were available from identical human embryonic face samples, namely CS18_12612, CS18_12695, CS19_12696, CS22_11963, and CS23_12492 (Supplementary Fig. 3, Supplementary Data 8). Mann–Whitney U test statistic was used to ascertain significance between the correlated enhancer:target gene pairs of interest versus all other pairs. We note that the correlation is highly significant but quantitatively moderate. This is likely due to technical factors, including imperfect enhancer-gene associations, target gene predictions not being available for all enhancers, differences arising from comparing predictions from human embryonic stem cells versus complex primary human embryonic tissue encompassing varying stages of differentiation, not excluding cases with redundant enhancers acting on the same gene(s), and uncertainty about the expected quantitative correlation between H3K27ac signal intensity at an enhancer and the expression level of a target gene. For the correlation for class "week-specific" in Supplementary Fig. 3b, the comparisons may not be significant due to the lack of capability of SRCC to detect patterns driven by one or two data points.

## GWAS data

The NHGRI-EBI Catalog of Genome-wide association studies[117] was mined for studies with the following keywords: craniofacial, face, cleft lip, cleft palate, microsomia, salivary, taste, and tooth. The compiled studies comprised diverse populations and ethnicities ranging from those belonging to the Unites States, Europe, Taiwan, China, Singapore, Korea and the Philippines, Brazil, Spain, Latin Americas, Uyghurs,

as well as admixed populations. For data published in the catalog by early 2022, we aggregated 41 studies representing normal facial variation as well as dento-oro-craniofacial disease. The SNiPA tool[118] was used for querying SNPs in LD ($r^2 \geq 0.8$) with the lead SNPs for the appropriate populations for the respective GWAS. This compilation of GWAS (Supplementary Data 9–10) was intersected with 13,983 reproducible human enhancers derived from primary embryonic bulk face between CS13-23. We have partitioned a total of 14,137,504 SNPs from the dbSNP155[119,120] catalog by their association with normal face variation or human disease and overlap with reproducible fetal human face enhancers described in this work. We found that 605 out of 27386 (2.3%) normal face variation- or human disease-associated SNPs overlapped the peaks, while only 245,727 out of 14,083,942 (1.8%) non-associated SNPs did. The overlap was significantly different from random expectation with an odds ratio of 1.27 (Pearson's Chi-squared test with Yates' continuity correction: X-squared = 34.102, $df = 1$, $p$ value = 5.229e-09).

### Intersecting VISTA catalog with predicted craniofacial enhancers

We intersected a subset of 130 human craniofacial regulatory elements (out of 3193 total curated) in the VISTA Enhancer Browser with 13,983 reproducible human candidate enhancers for weeks 4–8 from this study requiring a minimum 100 bp overlap (Supplementary Fig. 2, and Supplementary Data 4). We note that VISTA enhancers are not a random sample of the genome and are intentionally picked for their high levels of evolutionary conservation, high levels of epigenomic signal in embryos, lower repeat content, and proximity to genes known to regulate embryonic development.

### Single-cell RNA-seq

Both wild-type FVB/NJ crosses (ages 7–8 weeks), as well as transgenic mice harboring the *Hsp68* promoter and mCherry reporter at *H11* locus and generated as described earlier in Methods, were used. Transgenic embryos were harvested at the determined developmental stage, between 11.5–13.5 dpc (8 samples at e11.5, 1 sample at e12.5, and 4 samples at e13.5), and examined for positive *mCherry* signal if applicable. Embryos positive for *mCherry* reporter activity showed reproducible and comparable enhancer-reporter expression as seen in the *lacZ* expression patterns for VISTA enhancers hs1431, hs521 and hs746 used in this study. Embryos were consistently kept in ice-cold PBS until dissection. Upon fluorescence screening, developing face tissue was dissected with the aid of a Leica MZ16 microscope, and immediately processed for downstream experiments. Fresh mouse embryonic face tissue was mechanically dissociated by pipetting gently into a single-cell suspension using Accumax, assessed for viability of cells and for cell density using Trypan Blue staining. Individual cells were quantified, spiked with 10% HEK293T/17 frozen-thawed cells, and processed using the 10X Genomics Chromium Next GEM Single-Cell 3' protocol including transcript capture and library preparation for single-cell gene expression. Samples were either processed individually or pooled using a Multi-seq strategy[121] upstream of the 10X Genomics Chromium protocol. The resulting libraries were sequenced on an Illumina HiSeq2500 or NovaSeq 10X. BCL files from Illumina were processed into FASTQ format, individual sample libraries were de-multiplexed as necessary, reads were aligned to mm10 reference genome where *mCherry* sequence was added as an additional chromosome. Cell Ranger 3.1.0 software was used to process the raw sequence files and generate feature-barcode matrices. After correcting for batch effects, data from all libraries was aggregated into a single R object file using the 10X Genomics Cell Ranger 3.1.0. Seurat v3.2 guided clustering tutorial was used for formal downstream analyses[122–124]. Adhering to the standard pre-processing workflow and quality control, cells with unique feature counts>200 and <5%mitochondrial reads were retained. Based on the inspection of UMI/gene count plots, the UMI range, which preserved the main group of cells and excluded both droplet debris and likely clumps of cells was established for each sample separately (2000–4000 minimum, 15,000–60,000 maximum). For scRNA-seq, samples were integrated using standard Seurat procedure; *SelectIntegrationFeatures* function was run on a list of all 9 samples to be integrated to find 3000 most variable features. *mCherry* transcripts, genes on chromosomes X or Y (Gencode vM24) and cells expressing >5% mitochondrial genes (with names starting with *mt*) were removed from that list. *PrepSCTIntegration*, *FindIntegrationAnchors* and *IntegrateData* functions were run to obtain an integrated dataset. Normalization, feature selection, scaling, dimensional reduction, clustering and finding cluster biomarkers i.e., differentially expressed features were performed as guided. Our final Seurat/clustered UMAP consists of a 25,645 feature by 57,598 cell matrix, with a median of 1659 and a range of 500–8840 genes expressed per cell (Supplementary Fig. 5), and a range of 474–9,148 cells for the smallest to largest clusters (Supplementary Data 16).

### Assigning cell type identity to scRNA-seq clusters.

We systematically assigned cell type identities to the clusters in our craniofacial scRNA-seq dataset using two computational methods. (i) Using our primary single-cell dataset as query, we assigned cell type identities by Seurat-based automated reference mapping to a published large single-cell gene expression dataset[125] of whole mouse embryonic development for stages e9.5–13.5, the reference was downsampled to 100 K cells for efficient processing and retained all 38 broad cell types originally described. 27 cell types from the reference were summarily mapped in our craniofacial scRNA-seq dataset by Seurat's label transfer; the referenced cell types showed a good overall correlation with the cell types associated with the top 20 marker genes in most clusters in our *ScanFaceX* dataset. (ii) In parallel, we used the *scran's* scoreMarkers wrapper function described in the *scran* package, which uses effect sizes (Cohen's *d* statistic) to perform differential expression to list marker genes for each of the clusters in a scRNA-seq dataset[126,127]. These marker gene sets were tested for enrichment of Gene Ontology (GO) biological process terms by performing a hypergeometric test to identify GO terms overrepresented in our *ScanFaceX* dataset. Cell type annotations from methods (i) and (ii) described above were compared and resulted in each cluster in the *ScanFaceX* dataset having one or more cell type annotations. Finally, cell clusters that showed similar or close cell-type-specific signatures were manually merged to reflect 16 formal annotations for definitive cell types capturing craniofacial development and morphology. We note that the label "other craniofacial" encompasses a mix of cells with the following descriptive terms retained from the auto-referencing steps: palate development, roof of mouth, mesenchyme, and premature oligodendrocytes. (Supplementary Figs. 7, 9–11, Supplementary Data 14–16).

### Single-nucleus ATAC-seq

Wild-type FVB/NJ crosses (ages 7–8 weeks) were used to generate mouse embryos for each of the developmental stages e10.5–15.5. Face tissue was dissected, flash-frozen in liquid nitrogen (N2) and stored at −80 °C until ready to process. Tissue was transported to the Center for Epigenomics, University of California, San Diego School of Medicine, La Jolla, CA, for processing using a combinatorial indexing-assisted single-nucleus ATAC-seq strategy[77]. Briefly, nuclei were isolated and permeabilized in optimized conditions, pelleted and suspended in resuspended in 500 μL high salt tagmentation buffer. Nuclei were counted using a hemocytometer and 2000 nuclei were dispensed into each well of a 96-well plate per sample. A BenchSmart™ 96 (Mettler Toledo) was used to add 1 μL barcoded Tn5 transposomes to each of the wells in the 96-well plate, the mix was incubated for 60 min at 37 °C with shaking (500 rpm). EDTA at a final concentration of 20 mM was then added to each well for incubation at 37 °C for 15 min with shaking (500 rpm) to terminate the Tn5 reaction. Next, nuclei were suspended

in 20 µL of 2× sorting buffer (2 % BSA, 2 mM EDTA in PBS), wells for each sample were combined and stained with Draq7 at 1:150 dilution (Cell Signaling). 20 nuclei per sample were sorted per well into eight 96-well plates (total of 768 wells) in 10.5 µL of Elution Buffer (25 pmol primer i7, 25 pmol primer i5, 200 ng BSA (Sigma) using a Sony SH800. A Biomek i7 Automated Workstation (Beckman Coulter) was used for performing downstream steps. Samples were incubated at 55 °C for 7 min with shaking (500 rpm) in 1 µL 0.2% SDS, followed by addition of 12.5% Triton-X to quench the SDS. Samples were PCR-amplified (12.5 µL NEB Next High-Fidelity 2× NEB PCR Master Mix; [72 °C 5 min, 98 °C 30 s, (98 °C 10 s, 63 °C 30 s, 72 °C 60 s) × 12 cycles, held at 12 °C]). Wells were combined post-PCR. A manual MinElute PCR Purification Kit (Qiagen) along with a vacuum manifold (QIAvac 24 plus, Qiagen) was used for library purification, and size selection was performed with SPRISelect reagent (Beckmann Coulter, 0.55x and 1.5x). A Qubit fluorimeter (Life Technologies) was used to quantify the libraries, and the nucleosomal pattern of fragment size distribution was verified on a High Sensitivity D1000 Tapestation (Agilent). Libraries were sequenced on a NextSeq500 or HiSeq4000 (Illumina) using custom sequencing primers.

Reads were aligned to mm10 reference genome using bowtie2 with default parameters, and cell barcodes were added as a BX tag in the bam file. Only primary alignments were kept. Duplicated read pairs were removed with Picard, and proper read pairs with insert sizes less than 2000 were kept for further analysis.

**Clustering and cell type annotation.** snapATAC2 (version 1) package was used to perform read counting and cell clustering for both all-tissue clustering and tissue-level clustering[128]. First, we removed nuclei with less than 400 fragments or TSS enrichment <4 for all tissues and calculated a cell-by-bin matrix at 5000-bp resolution for every sample independently, binarized the matrices and subsequently merged them for each clustering task. Next, we filtered out any bins overlapping with ENCODE blacklist (mm10, http://mitra.stanford.edu/kundaje/akundaje/release/blacklists/mm10-mouse/mm10.blacklist.bed.gz). To stabilize the variance and reduce the impact of noise, we normalized the read coverage of all bins with log10 (count+1), applied Z-score transformation to ensure that each feature contributes equally to downstream analyses, and only removed bins with absolute Z scores higher than 2. After these filtering steps, we calculated Jaccard Index and performed dimensional reduction using the runDiffusionMaps function on similarity matrices. The memory usage of the matrices scales quadratically with the number of nuclei. Therefore, given the computational limitations at the time of analysis, and based on evidence provided by SnapATAC[128], we sampled a subset of 30,000 "landmark" nuclei to compute the matrices and then extended to the rest of the cells. After dimensional reduction, we selected the top 20 eigenvectors based on the variance explained by each eigenvector and computed 20 nearest neighbors for each nucleus and applied the Leiden algorithm (leiden clustering resolution = 1) to define 20 clusters.

To perform label transfer from the scRNA-seq to the corresponding snATAC-seq data we first created a gene activity matrix from the snATAC-seq data using accessibility in TSS and gene bodies with the SnapATAC package. We then converted our gene activity matrix into a Seurat object and used default parameters for the Seurat function *FindTransferAnchors* to perform canonical correlation analysis on the gene activity matrix along with the gene expression quantification from the scRNA-seq data. The *FindTransferAnchors* function in Seurat uses unsupervised identification of anchors representing cells from separate datasets, with the assumption that these cells are derived from shared biological states[129]. Finally, we used the *TransferData* function to annotate the snATAC-seq data via label transfer.

For the scatter plots showing normalized accessibility versus gene expression (Fig. 4b), we used a gene-by-cell matrix, which has counts for reads at the TSS and the gene body of each marker gene.

## Comparing human craniofacial enhancers with previously reported enhancer catalogs

We compared human enhancers identified in this study with a set of 5000 primate enhancers profiled from cranial neural crest cell differentiation using both chimpanzee and human cells and a list of 1000 human-biased enhancers[16]. Genomic coordinates of these enhancers were converted to hg38 using LiftOver and intersected with our list of 13,933 reproducible human enhancers. Similarly, enhancers identified by Cap Analysis of Gene Expression (CAGE) including those from normal human embryonic palatal mesenchyme (HEPM:CNhs11894) cells were obtained from the FANTOM5 database[97–99]. Genomic coordinates of the enhancer lists from FANTOM5 were converted to hg38 and intersected with the 13,983 human reproducible craniofacial enhancers from this study. Results of this analysis are reported in Supplementary Data 2.

## Imaging

For both brightfield and fluorescent images, all embryos were imaged with a Leica MZ16 microscope and a Leica DFC420 digital camera using identical lighting conditions.

## Statistics and reproducibility

Statistical analyses are described in detail in the Methods section above. For human embryonic face samples, we performed experiments with biological replicates as follows: three at CS18, one at CS19, two at CS22 (with two technical replicates for one of two samples), and one at CS19 for ChIP-seq. We performed experiments with two biological replicates at CS18, one each at CS19, and CS22-23 for ATAC-seq; four replicates at CS18, one at CS19, seven at CS22, and four at CS23 for RNA-seq. For single-cell experiments of the mouse face, we performed experiments for eight biological replicates at e11.5, and four replicates each at e12.5 and e13.5, respectively, for scRNA-seq, while single samples at each of the six mouse embryonic stages (e10.5, e11.5, e12.5, e13.5, e14.5, and e15.5) were processed for snATAC-seq. For transgenic assays primarily performed and reported in this study, we confirmed results in at least two independent animals (range 2-10 positive results) and used criteria consistent with our site-directed transgenesis pipeline established for the VISTA Enhancer Browser. Individuals who qualitatively assessed the results of in vivo transgenic reporter assays were blinded to genotyping information. For all other experiments, the investigators were not blinded to allocation during experiments and outcome assessment. No statistical method was used to pre-determine sample size. No data that passed quality control criteria for experiments were excluded from the analyses. The experiments were not randomized. Unless otherwise stated, default parameter settings were employed for any software tool that was used in the analyses. Whenever a p-value is reported in the text, the statistical test is also indicated. All statistics were estimated, and plots were generated using the statistical computing environment R/R version 4.1.0.

## Reporting summary

Further information on research design is available in the Nature Portfolio Reporting Summary linked to this article.

# Data availability

Wherever applicable, reference genomes Human GRCh38/hg38 and Mouse GRCm38/mm10 were used for alignment and comparisons. The ChIP-seq, ATAC-seq, RNA-seq, as well as scRNA-seq and snATAC-seq data presented in this publication and generated as part of this study are accessible at the National Institute of Dental and Craniofacial Research's FaceBase[79,108,130,131] Consortium (facebase.org), and can be

found under the following records: RNA-seq, ChIP-seq and ATAC-seq analysis of human fetal tissue, *FaceBase Consortium* Accession: FB00001358 [https://doi.org/10.25550/3C-4G62]. Single-cell RNA-seq and single-nucleus ATAC-seq analysis of mouse embryonic tissue, *FaceBase Consortium* Accession: FB00001359 [https://doi.org/10.25550/3C-4R98]. These data are additionally deposited in NCBI's Gene Expression Omnibus[132,133] and are accessible under GEO Series Accession GSE235858. Other published datasets used in the analyses are described in detail in Methods, cited studies[16,24,52,97,98,125] and listed in Supplementary Data. The NHGRI-EBI Catalog of Genome-wide association studies is accessible at https://www.ebi.ac.uk/gwas/home, and the FANTOM5 database is accessible at https://fantom.gsc.riken.jp/5/. Images of embryos with *lacZ*-reporter activity are available from the VISTA Enhancer Browser https://enhancer.lbl.gov/. Source data are provided with this paper.

## Code availability
No previously unreported custom computer code, mathematical algorithm or software was used in the analyses of data presented in this study. Current community-accepted and benchmarked bioinformatic methods were used and are appropriately cited in the main text and Methods.

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

## Acknowledgements

This work was supported by US National Institutes of Health (NIH) grants (R01DE028599 and U01DE024427) to A.V. Research was conducted at the E.O. Lawrence Berkeley National Laboratory and performed under US Department of Energy Contract DE-AC02-05CH11231, University of California (UC). The human embryonic and fetal material was provided by the Joint MRC/Wellcome (MR/R006237/1, MR/X008304/1, and 226202/Z/22/Z) Human Developmental Biology Resource (www.hdbr.org). F.D. was supported by the Swiss National Science Foundation grant P400PB_194334. M.O. was supported by the Swiss National Science Foundation (SNSF) grant PCEFP3_186993. We thank Yoon Gi "Justin" Choi of the University of California, Berkeley QB3 Genomics Core for technical assistance with the 10X Genomics experimental set-up.

## Author contributions

S.S.R., D.E.D., L.A.P. and A.V. designed the study. S.L. coordinated and oversaw the collection of samples at the HDBR site. S.S.R., C.S., M.O., Y.Zhu, H.W., S.Y.A., J.A.A., V.A., S.T., I.P-F., C.S.N., M.Kato, R.H., K.V.M., A.W., L.L., S.P. performed experiments. J.A.A. performed imaging. S.S.R., K.P., M.L.A., M. Kosicki, L.E.C., F.D., M.B., G.K., I.B., Y.F.-Y. analyzed data. B.R. supervised snATAC-seq experiments and integrative analysis of snATAC-seq and scRNA-seq data. S.S.R., L.P., and A.V. wrote the manuscript with input from the remaining authors.

## Competing interests

Bing Ren is a co-founder of Arima Genomics, Inc, and Epigenome Technologies, Inc. The remaining authors declare no competing interests.

## Additional information

[1]Environmental Genomics & System Biology Division, Lawrence Berkeley National Laboratory, 1 Cyclotron Road, Berkeley, CA 94720, USA. [2]Bioinformatics and Systems Biology Graduate Program, University of California San Diego, La Jolla, CA, USA. [3]U.S. Department of Energy Joint Genome Institute, 1 Cyclotron Road, Berkeley, CA 94720, USA. [4]Center for Epigenomics, University of California San Diego School of Medicine, La Jolla, CA, USA. [5]Institute of Experimental and Clinical Pharmacology and Toxicology, Faculty of Medicine, University of Freiburg, Freiburg, Germany. [6]Biosciences Institute, Faculty of Medical Sciences, Newcastle University, Newcastle NE1 3BZ, UK. [7]Institute of Genome Medicine, Moores Cancer Center, University of California, San Diego School of Medicine, La Jolla, CA, USA. [8]Comparative Biochemistry Program, University of California, Berkeley, CA 94720, USA. [9]School of Natural Sciences, University of California, Merced, CA, USA. [10]Present address: Department of Genetic Medicine and Development, Faculty of Medicine, University of Geneva, 1211 Geneva, Switzerland. [11]Present address: Department for BioMedical Research (DBMR), University of Bern, 3008 Bern, Switzerland. [12]Present address: Department of Cardiology, Bern University Hospital, Bern 3010, Switzerland. [13]Present address: Lucile Packard Children's Hospital, Stanford University, Stanford, CA 94304, USA. [14]Present address: The Jerusalem Center for Personalized Computational Medicine, Hebrew University of Jerusalem, Jerusalem, Israel. [15]Present address: Center for Cancer Research, Medical University of Vienna, Borschkegasse 8a 1090, Vienna, Austria. [16]Present address: Department of Surgery and Cancer, Imperial College London, London, UK. [17]Present address: University Research Management Center, Tohoku University, Sendai, Miyagi 980-8577, Japan. [18]Present address: UC San Francisco, Division of Experimental Medicine, 1001 Potrero Ave, San Francisco, CA 94110, USA. [19]Present address: Octant Inc., Emeryville, CA 94608, USA. ✉e-mail: avisel@lbl.gov

