## [Peer Review File · Nature Communications]

Dynamic Enhancer Landscapes in Human Craniofacial DevelopmentREVIEWER COMMENTS

Reviewer #1 (Remarks to the Author):

In this paper, the authors generated H3K27ac ChIPseq data of Carnegie stages 18-23 to expand an existing resource of enhancer candidates involved in human craniofacial development (Wilderman et al., Cell Rep, 2018).

As a support for the identified candidates, peak regions were overlapped with ATACseq data and functional enhancers from the VISTA database. A subset of the human enhancers was validated by transgenic reporter assays in mouse and GO term enrichment analysis was performed on genes linked to the enhancers by distance.

The authors show that a majority of the identified enhancer candidates have conserved sequences and chromatin features in mice. In the mouse, the authors then provide a cell-type resolved resource of craniofacial enhancers, defining them by promoter-distal single cell accessibility and use reporter assays in combination with scRNAseq to possibly predict cell-type specific activities of enhancers. Nonetheless, the evidence provided by the authors regarding this aspect is insufficient (see below), also considering that a given enhancer can be active in several cell-types or tissues. In this respect, identifying the target genes of relevant enhancers would have been of broader interest. However, this analysis is missing from the study. Identification of enhancers that could mechanistically be linked to facial variation or craniofacial birth defects is also missing. Despite the scientific interest of this impressive catalogue of data, the present study may not provide sufficient novelty for publication in Nature Communications. At any rate, this study requires extensive revision before publication.

Major points :

1. To investigate the temporal dynamics of human craniofacial enhancers, correlation between H3K27ac ChIP-seq data and ATAC-seq data should not be analyzed by merging all developmental stages (as it seems to be) but should be stage-specific.

Indeed, supplementary table 3 shows H3K27ac-bound regions (ChIP-seq) from human developing face between weeks 4-8 of embryogenesis that overlap with open chromatin regions from corresponding face tissue, but no indication is given of the stage(s) at which these correlations occur, knowing that for stages CS13 to CS17, ChIP-seq data come from Wilderman et al., 2018, where no ATAC-seq data were generated, and that for stages CS19 and CS23, a bulk ATAC-seq experiment was performed (at each stage) but no ChIP-seq.

In the method section, it is reported that the H3K27ac ChIP-seq peaks from all stages of development (CS13 to CS17) were merged, resulting in 13,983 reproducible human candidate enhancers. Line 97 – 107: it seems that this pool of H3K27ac peaks was correlated with the ATAC-seq peaks that were also merged from all stages (with the exception of stages CS13 to CS17, where they were not available), to look if a correlation was present. If so, this means that the correlation between the H3K27ac signal and chromatin accessibility is not stage-specific, which is problematic, specifically because the authors identified several H3K27ac-positive elements that were not accessible, questioning their role as active enhancers. To identify enhancers that are active at each developmental stage, it would have been more accurate to correlate the H3K27ac-bound regions (ChIP-seq) with ATAC-seq signals at each developmental stage.

Similarly, when the authors explore the temporal dynamics of human craniofacial enhancers

(lines 178 to 192), it would be preferable to use both H3K27ac-bound regions (ChIP-seq) and ATAC-seq signals to identify active enhancers, instead of H3K27ac ChIP-seq alone.

2. Moreover, Lines 101-102 and Figure 2b: I have doubt that the conclusions based on temporal resolution are valid. Are the two integrated datasets (Wilderman et al. and this study) comparable (different H3K27ac antibodies were used, sequencing depth, protocol differences, MACS2 parameters)? This may cause systematic differences in how enhancers are identified in the two separate datasets, reducing the expected overlap in different stages without real temporal differences in enhancer activity.

3. Lines 182-186 and figure 2b: The numbers do not match. 1,624 continuously active enhancers + 3,137 continuously active enhancers in a subset of timepoints do not sum up to 5,373 as indicated in the figure. In figure 5b, it is not clear where are the 1,624 elements predicted to be continuously active as enhancers throughout all five weeks (line 182 – 184), and where are the 3,137 elements that showed continuous activity periods covering a subset of the five weeks (lines 185-186). Instead, we observe a block of 5,373 ‘continuous elements’, which does not correspond to the sum of 1,624 + 3,137. Moreover, these data should be presented in a table, containing the coordinates of the active regulatory elements plus their week of activity.

4. Line 143: how are defined the presumptive target genes of the human candidate craniofacial enhancers identified by the authors? The authors identified the nearest genes to specific enhancers as potential target genes for those enhancers. However, only a fraction of enhancers are known to target their nearest genes, as enhancers can show selective directed interactions with far away genes, often skipping proximal promoters. This raises the possibility that other mechanisms, such as long-range chromatin interactions, may regulate the target gene-promoter interactions for the remaining enhancers.

5. Lines 205-220. To assess whether the differences in epigenomic signatures between human and mouse translate into species-specific differences in in vivo enhancer activity, the authors used a transgenic mouse assay to compare the human and mouse orthologs of a predicted human-specific enhancer. They found that ‘embryos transgenic for the human ortholog (hs2656) show reproducible activity in the developing nasal and maxillary processes at embryonic day (e) 12.5, confirming that the human tissue-derived enhancer signature correctly predicts in vivo activity at the corresponding stage of mouse development (Figure 2c). In contrast, we did not observe reproducible craniofacial enhancer activity with the mouse orthologous sequence, concordant with the absence of enhancer chromatin marks in mouse at this location.’

It should however be mentioned that hs2656, while being conserved between human and mouse is not identical between these 2 species, thus probably explaining the differences in epigenomic signatures between mouse and human. What are the differences in the orthologs (human and mouse) that could potentially create differences in the in vivo assay when tested in the mouse system? If not in the result section, some explanation/ speculation in the discussion would be illuminating for the reader (see also below).

6. Lines 131-134: 44 human enhancers could not be validated in the mouse despite their H3K27ac signature. Please comment on this in the light of the claim in lines 216-217 and figure 2c/d. From one example you cannot conclude that “human tissue-derived enhancer signature correctly predicts in vivo activity at the corresponding stage of mouse development” (lines 216-217). Which of these human enhancers are positive for H3K27ac after integration into the mouse genome? Additionally, the sequence conservation between

hs2656 and mm2289 is not perfect and may explain differences in H3K27ac and functionality (see above). Therefore, the conceptual distinction between sequence and epigenetic conservation is elusive.

7. Lines 265 – 267 “we observed 16,564 differential accessible regions (DARs) across 20 separate clusters, indicating that each of the clusters representing identical or very similar cell types have distinct open chromatin signatures”. This sentence is not clear. How can the authors claim that the clusters identified by snATAC-seq represent identical or very similar cell types? Knowing that all tissues from the craniofacial prominences (i.e. epithelium, endothelium, mesoderm, neural crest cells...) were included in the analysis.

8. The authors claim that the putative enhancer regions near *Isl2* and *Scaper* (mm2285), and in the intron of *Lrrk1* (mm2282) drive reproducible lacZ-reporter activity in the developing mouse face in anatomical regions that are consistent with neuronal and epithelial cell types. This statement should however be confirmed by doing a double staining on sections (LacZ + a neuronal marker for mm2285 / LacZ + an epithelial marker for mm2282). Moreover, without knowing the target genes of these enhancers, it is hard to predict where the LacZ staining should be. The authors make the supposition that the target genes are the nearest ones, but it is not necessarily the case, it is even often wrong (see above). Along this line, we can appreciate that mm2285 does not reproduce the pattern of expression of *Isl2* or *Scaper*, and that mm2282 does not reproduce the pattern of expression of *Aldh1a3*. The authors should then first determine the target gene of the enhancer, check that the LacZ reproduce the expression pattern or part of the expression pattern of this target gene, check in which cell type(s) this target gene is expressed, and check in their snATACseq data that their enhancer is indeed open in the cell type(s) where the LacZ is present.

9. Lines 316 to 331. To validate their cell type-resolved enhancer predictions, the authors intersected their data with craniofacial enhancers validated in vivo and curated in the VISTA Enhancer Browser. However, as mentioned above, it's not possible to determine the cell type specificity of the VISTA enhancers, only based on a LacZ staining performed on whole mount embryos. For instance, the authors claim ‘the chondrocyte cluster (cluster 13) has multiple VISTA enhancers with activity in the mid-face, paranasal regions, and/or the region at the junction of the developing forebrain and nasal prominences that are consistent with developing cartilaginous regions of the face’. Looking at regions where an enhancer is active is not sufficient to make conclusions on the cell-type specificity of this enhancer. The last part of the analysis performed by the authors, called “Cell Type-specific Enhancer Activity at Single-cell Resolution” (from line 340) indeed explores the cell type specificity of selected craniofacial enhancers in vivo, but this analysis is performed on only 3 enhancers and none of them are specifically active in one cell type (presence of multiple red points for each of the enhancer tested, figure 6c). Of note hs746 is not only active in osteoblasts as claimed in the text, but also in the eye (as seen on the figure 6a and 6b). Similarly, hs521 is not only active in a subset of mesenchymal cells and chondrocytes progenitors, as claimed in the text, but also in the eye (as seen on the figure 6a and 6b).

10. Line 99: The RNAseq data introduced here is not mentioned anywhere later in the manuscript. It is also listed in supplementary table 1 though. Please specify why it was included in the manuscript and report the results. The RNAseq data could be used to correlate expression and H3K27ac levels across stages, for example. This would be particularly interesting for the predicted stage-specific enhancers. Please check and report whether the corresponding genes are also generally expressed at lower levels in that specific stage.

11.

a) Lines 101-102: Please provide more details on how the data from Wilderman et al. was integrated into your analyses. For example, which regions from Wilderman et al. were used (ChromHMM predicted enhancers or H3K27ac peaks)?

b) Similarly, please also specify what was considered “an epigenomic enhancer signature in the mouse” (lines 199-200) which is currently unclear.

c) Additionally, in line 105 the number of enhancers in later stages (CS18-23) is mentioned (10,893 regions, data generated in this study). It is left unclear whether there are only ~3,000 enhancers found in earlier stages (Wilderman et al. data) due to generally lower numbers of enhancers identified by Wilderman et al.

d) Roughly, the total number of enhancers per week seems different (figure 2b) with less enhancers found in the stages/weeks only covered in the Wilderman et al. dataset. Technical differences are likely reinforced by varying number of replicates for different weeks. Consequently, the limitations mentioned in lines 188-189 apply to all groups of enhancers (week-specific, continuous, non-continuous). Therefore, please adapt and expand this statement on the limitations with respect to the temporal dynamics of enhancer activities.

12. Line 497: Did the peaks have to be present in at least 2 replicates of each week? This would mean that no week-specific peaks can be identified, which is in contrast with the results of figure 2b. In case they were only required to be present in at least two replicates of one, i. e. the same, week, the term “highly reproducible” (l. 543, l. 152) seems to be an overstatement. Please clarify.

13. Line 125: How much overlap of the VISTA enhancers and the enhancers identified in this study set would you expect to see by chance? This is considering general mappability, accessibility, etc. of the genome.

14. Line 132: How were the 60 enhancer candidates selected? Please also add a statement to lines 133-134 about how many of the 60 tested enhancers showed activity in other structures than the face.

15. Line 144: The rGREAT analysis is missing from the methods section. How were target genes assigned to enhancers? For how many enhancers could you identify a target gene? Against which background was ontology enrichment performed?

16. Lines 171-177: It is unclear how this analysis supports the identified enhancers, since only LD SNPs that are inherited together with the putative causal SNPs overlap with enhancer candidates and no causal SNPs itself.

Line 175: Is there an enrichment of LD SNPs in the enhancers or is it just an overlap you may see by chance?

17. Fig 3 and 4: Why did the authors analyze different stages of embryonic development for scRNAseq and snATACseq? Do the conclusions still hold for the subset of snATACseq from only E11.5-13.5 (corresponding to the stages used in scRNAseq)? Please include results from this analysis of a subset of the data. Why are replicate numbers from the stages different?

18. Figure 4b: The authors picked the TSS with highest correlation between ATACseq and RNAseq data to investigate correlation between the data modalities, which will bias the analysis. Please consider picking the TSS with the highest accessibility instead.

19. Figure 4b: What does the width of the bars indicate? And why do they have the same height?

20. In figures 1b, 2c, 2d phyloP scores are shown, while placental conservation is depicted in figure 4c. Please describe which phyloP score was used and why you switched to placental conservation in figure 4c.

21. Line 270: The authors describe the number of DARs retained, but not the number of clusters out of the 16 annotated cell types from ScanFaceX that could be retained or the percentage of single cells from the ATACseq data that could be assigned to a cell type.

22. Line 322: Are these 77 VISTA confirmed snATACseq enhancers already among the previously identified 153 H3K27ac-enhancers (line 125) or different ones? It may simplify the readers' understanding to include a summary figure or table for numbers of enhancers and corresponding overlaps from: Wilderman et al., H3K27ac enhancers from this study, VISTA, snATAC from this study after liftOver

23. Please expand the discussion section. Add comments on limitations (identification of enhancers through other methods and validation, assignment of target genes, integration into H11 locus which takes the genes and enhancers out of their 3D context, etc.). Discuss transferability between humans and mice, too.

Additional methodological points:

1. QC Steps: While it is implied that quality control (QC) steps were performed during the analysis, it would be beneficial to explicitly mention the QC criteria used and any additional steps taken to ensure data quality and reliability.

2. Reproducibility: It's essential to mention the versions of software/tools used in the analysis, as well as any specific parameters or configurations applied.

3. Adding Specific Metrics: Some key quality metrics (e.g., the number of cells, median genes per cell, mitochondrial percentage, etc.) after each preprocessing step could provide a better understanding of the data's quality and how filtering was applied.

4. Batch Correction Method: It is mentioned that batch effects were corrected, but the specific method used for batch correction should be stated explicitly.

5. Visualization of QC Results: Including a few plots or summary statistics related to the QC steps (e.g., gene count distributions, mitochondrial percentage distributions). The analysis and methods described for clustering and cell-type annotation using the snapATAC package appear well thought out and comprehensive. However, there are a few suggestions for clarity and potential improvements:

a) Method Parameters: For reproducibility, authors should consider providing specific parameter settings used for the snapATAC package and the Leiden algorithm for clustering.

Mentioning the versions of software/tools used is also helpful.

b) Explain Data Transformation: While it is mentioned that the read coverage of bins was normalized with $\log_{10}(\text{count}+1)$ and Z-score transformation, a brief explanation of why these transformations were chosen and how they benefit the analysis could be included.

c) Data Subset Choice: Explain the rationale behind choosing 30,000 "landmark" nuclei for the dimensional reduction step and clarify how this subset was selected to ensure it is representative of the overall dataset.

d) Annotation Transfer: Describe the rationale behind using the default parameters for the FindTransferAnchors function in Seurat. If there were any manual adjustments made, it would be helpful to mention them.

Minor points:

1) Figure legend of figure 1: The abbreviation PA2 is not explained. Please add pa2 to the list of abbreviations in the figure legend.

2) Graphical abstract, figure 1a, figure 2c: Depiction of human and mouse - why is the yellow colour used for the human figure? We also suggest removing the background colors in figures 2c and 2d.

3) Extended data figure 1: What do the asterisks and the purple color stand for? Please explain in the figure legend.

4) Figure legends of figures 4b and 4c: We suggest adding information that colours in figures 4b and 4c correspond to the colour code used in figure 4a.

5) Figure 4c: The LacZ staining for the intronic enhancer is barely visible. You may indicate it with arrows, for example.

6) Suppl. table 1, row 6: Sample name (column 1) includes CS19, while stage (column 4) says CS18. This is contradictory. Please correct.

7) Suppl. table 2: Please add an additional column to indicate in which weeks and/or Carnegie stages the respective H3K27ac peak was seen.

8) Suppl. table 13: The header of column 7 contains "annoated". Please correct typo.

9) Suppl. table 1: The header of the table contains "detaisl". Please correct typo.

10) Suppl. table 5: There is one single entry with stage e12.5. Is this correct or was this maybe unintentionally added e. g. by Excel's autocompletion?

11) Line 35: This seems to be an overstatement, since a fraction of the enhancers was already identified before by Wilderman et al. Please modify.

12) Line 43, line. 200: Functional conservation is claimed based on the presence of histone marks, which do not prove functionality.

- 13) Line 197: How did you define “alignable orthologous sequence”? Please specify and provide more details on this.
- 14) Line 199: Please indicate that the percentage is in relation to the total number of human enhancer candidates (and not to the mouse enhancers).
- 15) Line 266: Please remove “representing identical or very similar cell types”. Up to this point in the text, you haven’t shown yet how the snATACseq clusters translate to cell types.
- 16) Line 492, line 520: Why did you use hg19 and hg38 for different analyses?
- 17) Line 494: Please provide more details on identification of overlapping peaks. `overlap_peaks.py` seems to take only 2 input files at a time. How did you handle this with your replicates? Please expand methods section accordingly.
- 18) Line 511: Have you performed all steps of the ENCODE ATACseq pipeline, including IDR, pseudoreplicated peaks, etc? Please provide details.
- 19) As an additional validation for the identified enhancers, could you please check and report the overlap with the FANTOM5 enhancer atlas, specifically from the HEPM cell line?
- 20) Lines 481-482: The word “with” is used twice next to each other. Please remove one occurrence.
- 21) Lines 124 to 129, instead of indicating the total number of regulatory elements that have been tested to date in both human and mouse in VISTA (3193 mentioned in the paper, 3320 when this reviewer checked on the site), it would be more appropriate to mention only the craniofacial regulatory elements, and only in Human.
If in VISTA I select the same criteria as the authors did in the supplementary figure 1 to select for the craniofacial enhancers (i.e. ear, branchial arch, eye, melanocytes, nose, cranial nerve, facial mesenchyme, mesenchyme derived from neural crest, trigeminal V (ganglion, cranial)) and in human, I found 254 elements. It means that among these 254 human craniofacial regulatory elements, they have identified 153 cases with their present human-derived epigenomic dataset.
- 22) Supplementary figure 4 and extended data figure 1: it is not specified from which species(s) the 506 craniofacial enhancers were selected in the VISTA enhancer browser (mouse? human? both?). It seems that they were selected from both human and mouse. As the authors' aim was to compare the human craniofacial enhancers they identified with the enhancer’s reporter in the VISTA enhancer browser, it would be more appropriate to select only the human enhancers in VISTA.
- 23) Supplementary table 5: the meaning of abbreviations is missing.
- 24) Extended data figure 3: the gene names (left side) are not visible.
- 25) Extended data figure 5: (top) the names of the cell types are not visible.

Reviewer #2 (Remarks to the Author):

This study performs an impressive amount of bulk RNA, ATAC, and H3K27Ac sequencing of human embryonic craniofacial tissues at various stages, as well as single-cell RNA and ATAC sequencing of mouse embryonic craniofacial tissues. It clearly provides an important and extensive resource for researchers interested in the gene regulatory basis of human craniofacial variation and disease. Another strength is the extensive validation of human enhancers in mouse, with 1/4 of those tested showing craniofacial expression. A large amount of new and published datasets, including extensive GWAS data, are integrated, and all datasets are publicly available to the community. Before publication, some issues with data interpretation should be addressed. The cell type annotations in places appear inaccurate, such as the cluster referred to as osteoblasts. This may reflect reliance on automated cell cluster calling rather than manual annotation based on the large number of single-cell studies that have already been performed of the craniofacial complex. A general weakness of the paper is the lack of citations of single-cell studies published by other groups, including similar studies that have performed integrated single-cell RNA and ATAC sequencing of multi-stage craniofacial tissues in other vertebrates. There is also concern that some of the conclusions have been over-interpreted, in particular the ability of *in silico* data to predict the activities of the enhancers tested (since only a minority confirm predictions). Nonetheless, provided these issues can be addressed, this study will represent a significant resource for the craniofacial community.

1. Annotation of the mesenchymal cell types in Fig. 3 appears inaccurate. Why is *Col2a1* described to label chondrocyte progenitors and not chondrocytes? In Extended Data Fig 3 there is a clear chondrocyte cluster marked by *Col2a1*, *Col9a1*, *Sox9*, etc. More significantly, the cells marked by *Msx1* do not appear to be osteoblasts as stated. The cluster marked by *Msx1* co-expresses *Prrx1*, *Trps1*, *Tfap2b* which are markers of perichondrium/undifferentiated mesenchyme and not osteoblasts. Better markers of osteoblasts include *Ifitm5*, *Sp7*, *Ocn* but these do not appear in the top 10 list of any cluster. There is also no clear osteoblast cluster in Extended Data Fig 5. One likely explanation is that osteoblasts were not captured in the scRNAseq experiment. Osteoblasts can be difficult to release from mineralized matrix during dissociation and are low or not represented in other single-cell studies of the head. Lastly, what is meant by "skeletal_other"? Teeth, tendon, something else? General annotations of the mesenchyme clusters lack sophistication and would benefit by comparing to experimentally validation annotations in previous single-cell craniofacial studies.

2. Line 76 and elsewhere should also cite the many single-cell studies already performed for mouse and zebrafish craniofacial tissues. In particular, Fabian et al, 2022 characterized craniofacial enhancers at multiple embryonic and adult stages with single-cell RNA and ATAC and showed multiple examples of cell- and stage-specific craniofacial enhancers.

3. Contrary to what is stated, Extended Data Fig. 9 show little correlation between scRNAseq and snATACseq data for more than half of the clusters, including "osteoblasts".

4. Authors should comment on why only 16/60 candidate human enhancers showed craniofacial expression in mouse transgenics, as well as why 13/60 showed expression in other regions but not in the face. What does it mean that 56% are functionally conserved (from Introduction) since only 16/60 human enhancers had craniofacial activity in mouse transgenic assays? Could there be technical reasons why only 1/4 of human craniofacial

enhancers had the expected activity in mouse? For example, it is known that different enhancers have different preferences for minimal promoters.

5. Section starting line 140 - how enhancers were linked to nearby genes needs to be better described. Did this leverage combined chromatin/RNA datasets?

6. More data should be provided for the 16 validated human enhancers - how close was their stage- and cell type-specific expression to that predicted by chromatin profiling? In Fig. 5b, it is not convincing that cluster 13 enhancers label chondrocytes. The level of resolution is insufficient to make this claim and many of the patterns look similar to those described for the cluster 2 connective tissue enhancers. For example (but not limited to this example), mm901 appears to label the near entirety of the first and second branchial arches, inconsistent with a specific chondrocyte expression pattern. From the images provided, individual cells cannot be resolved. In Extended Data Fig 11, it is unclear from the resolution provided whether mm2281 labels muscle as predicted. For at least a few cases, co-labeling of mCherry/lacZ with a cell type-specific marker (e.g. Sox9) would help resolve the cell type in which enhancers are active and how this confirms/denies in silico predictions.

7. Conclusions from Fig. 6 are over-stated. The scRNAseq analysis of mCherry+ cells only weakly at best corroborates the cell type predictions. The Msx1 and Gbx2 enhancers are most enriched in the "eye, melanocytes" cluster, and Gbx2 mCherry is otherwise very broad in many clusters including "other cellular". While the mCherry feature plots in Fig. 6a do look close to the RNA feature plots for the corresponding genes in Fig. 6c (strong data), the dot-plots are confusing.

8. The Discussion is very short and could be expanded to discuss caveats/limitations of the data. For example, Lines 200-201 - can the authors speculate on why 28% of human enhancers sequence conserved in mouse yet mouse enhancers do not show evidence of enhancer activity based on H3K27Ac?

9. Line 170: how relevant are some of the studies in Supp. Table 6, such as those involved in bitter/sweet/etc taste sensation?

Minor Comments

1. Bottom of graphical abstract, confusing to have UMAPs for cell types and enhancer activity flipped
2. Fig. 2C,D - unclear where hs2656 is located in tracks.
3. No Author Contribution listed for Bing Ren.

Reviewer #3 (Remarks to the Author):

In this study, the authors provide a large dataset of regulatory regions from the developing human face, identifying ~14,000 enhancers from the facial region at embryonic weeks 4 to 8. Using a transgenic mouse reporter assay, they validated numerous putative enhancers that were conserved in facial regions between mouse and human. Using bioinformatics analysis

to integrate the human data with single cell RNA-seq and single nuclei-ATAC seq of mouse craniofacial tissue, they speculate that 56% of these enhancers are conserved between mouse and human. Altogether, the work is nicely done and provides a large and useful resource for the community.

While the work seems highly appropriate for Nature Communications, I would recommend modifications as described below.

1. The discussion is very short and does not do a particularly good job of putting the present work in context of the existing literature on this topic. In particular, the authors should comment on overlap or lack thereof between the enhancers found in the present study compared with Prescott et al., 2015 and Wilderman et al., 2018.

2. Apropos of the above, it would be very useful for the authors to directly compare their 14,000 enhancers to those identified in previous studies and by Vista. How many are novel other than the few described in the present paper? How many were previously described using older technology?

3. On lines 381- 383, the authors state “that while many enhancers are functionally conserved between human and mouse, additional human-specific enhancers that show no functional conservation in mice can be identified by profiling human tissues.” They conclude that these are “human specific”. However, this cannot be concluded without functionally testing these enhancers in both human and mouse cells. Putative human enhancers that fail to provide a signal when tested in mouse are not necessarilyt "human specific", as this is a negative result.

To demonstrate a positive results, i.e. that these enhancers are human specific, the authors must test those that fail to express in mice in a human system. A simple way to do this would be to differentiate human ES cells into cranial neural crest cells (as done by Prescott et al., 2015) and test the ability of enhancers to mediate reporter expression in vitro. Only when they are positive in humans and negative in mouse at all stages can one conclude specificity.

4. The methods for obtaining human facial tissue are inscrutable and therefore difficult to evaluate. The authors need to specify what tissue was dissected at each stage and how this was done.

Point-by-Point Response to Reviewers' Comments

NCOMMS-23-28972: Cell Type- and Tissue-specific Enhancers in Craniofacial Development by *Rajderkar et al.*,

We thank the reviewers for their comments. Below we have summarized our responses to the points raised and changes to the manuscript.

Reviewer #1 (Remarks to the Author)

In this paper, the authors generated H3K27ac ChIPseq data of Carnegie stages 18-23 to expand an existing resource of enhancer candidates involved in human craniofacial development (Wilderman et al., Cell Rep, 2018). As a support for the identified candidates, peak regions were overlapped with ATACseq data and functional enhancers from the VISTA database. A subset of the human enhancers was validated by transgenic reporter assays in mouse and GO term enrichment analysis was performed on genes linked to the enhancers by distance. The authors show that a majority of the identified enhancer candidates have conserved sequences and chromatin features in mice. In the mouse, the authors then provide a cell-type resolved resource of craniofacial enhancers, defining them by promoter-distal single cell accessibility and use reporter assays in combination with scRNAseq to possibly predict cell-type specific activities of enhancers. Nonetheless, the evidence provided by the authors regarding this aspect is insufficient (see below), also considering that a given enhancer can be active in several cell-types or tissues. In this respect, identifying the target genes of relevant enhancers would have been of broader interest. However, this analysis is missing from the study. Identification of enhancers that could mechanistically be linked to facial variation or craniofacial birth defects is also missing. Despite the scientific interest of this impressive catalogue of data, the present study may not provide sufficient novelty for publication in Nature Communications. At any rate, this study requires extensive revision before publication.

We appreciate the candid feedback and thank the reviewer for their constructive and valuable suggestions for the improvement of the manuscript, which are described in detail below.

Major points:

1. To investigate the temporal dynamics of human craniofacial enhancers, correlation between H3K27ac ChIP-seq data and ATAC-seq data should not be analyzed by merging all developmental stages (as it seems to be) but should be stage-specific. Indeed, supplementary table 3 shows H3K27ac-bound regions (ChIP-seq) from human developing face between weeks 4-8 of embryogenesis that overlap with open chromatin regions from corresponding face tissue, but no indication is given of the stage(s) at which these correlations occur, knowing that for stages CS13 to CS17, ChIP-seq data come from Wilderman et al., 2018, where no ATAC-seq data were generated, and that for stages CS19 and CS23, a bulk ATAC-seq experiment was performed (at each stage) but no ChIP-seq. In the method section, it is reported that the H3K27ac ChIP-seq peaks from all stages of development (CS13 to CS17) were merged, resulting in 13,983 reproducible human candidate enhancers. Line 97 – 107: it seems that this pool of H3K27ac peaks was correlated with the ATAC-seq peaks that were also merged from all stages (with the

exception of stages CS13 to CS17, where they were not available), to look if a correlation was present. If so, this means that the correlation between the H3K27ac signal and chromatin accessibility is not stage-specific, which is problematic, specifically because the authors identified several H3K27ac-positive elements that were not accessible, questioning their role as active enhancers. To identify enhancers that are active at each developmental stage, it would have been more accurate to correlate the H3K27ac-bound regions (ChIP-seq) with ATAC-seq signals at each developmental stage.

We apologize for the potentially confusing description of this analysis. To investigate the correlation between H3K27ac and ATAC signal, we only used timepoints (CS18-23) for which we have both H3K27ac and ATAC data (as suggested by the reviewer). H3K27ac input into this analysis is 10,893 CS18-23 out of total 13,983 CS13-23 peaks, of which 6,718 (61.7%) also showed accessible chromatin signal.

We have followed the reviewer's suggestion and examined this in more detail focusing on week 7, which is the stage for which we have the largest number of perfectly matched datasets, i.e., ChIP-seq and ATAC-seq derived from the same biological samples. We examined three biological samples that were processed for both H3K27ac ChIP-seq and ATAC-seq (CS18-12676, CS18-12695, CS19-12696). We observed that 70% (2225/3182) of reproducible H3K27ac peaks (defined by presence in at least two samples) overlap at least one ATAC-seq peak derived from the same samples. We have now specifically included this analysis in the main text as follows:

pg 6, lines 113-118, Results

We examined the correlation between H3K27ac peaks and chromatin accessibility focusing on week 7 (comprising CS18 and CS19), since the largest number of perfectly matched datasets (H3K27ac peaks and chromatin accessibility data from the same biological samples) were available for this stage. We observed that 2,225 out of 3,182 (70%) of the reproducible H3K27ac peaks overlap at least one ATAC-seq peak derived from the same samples (**Supplemental Table 3; Methods**).

Similarly, when the authors explore the temporal dynamics of human craniofacial enhancers (lines 178 to 192), it would be preferable to use both H3K27ac-bound regions (ChIP-seq) and ATAC-seq signals to identify active enhancers, instead of H3K27ac ChIP-seq alone.

Since we have matching H3K27ac-ATAC data only for a subset of the later embryonic stages, we chose to conduct this analysis using only H3K27ac data from weeks 4-8 to ensure a uniform and consistent analysis approach across the entire developmental window covered by our analysis.

2. Moreover, Lines 101-102 and Figure 2b: I have doubt that the conclusions based on temporal resolution are valid. Are the two integrated datasets (Wilderman et al. and this study) comparable (different H3K27ac antibodies were used, sequencing depth, protocol differences, MACS2 parameters)? This may cause systematic differences in how enhancers are identified in the two

separate datasets, reducing the expected overlap in different stages without real temporal differences in enhancer activity.

For the datasets from Wilderman et al., 2018, we used publicly available and post-mapped TagAlign files. Considering these data sets were generated by different labs, using different protocols (including a different antibody for H3K27ac), it is critical to carefully assess for systematic biases and batch effects. We also note that an additional source of variation stems from the use of human embryonic tissues, which are typically derived from individual or a small number of fetal tissue donations, which can result in minor variation regarding tissue dissection and inevitably introduces genetic heterogeneity as compared to work with tissues from isogenic mouse strains. Recognizing these unavoidable sources of variation and potential batch effects, we aimed to make the analysis as comparable as possible, by down sampling the number of input reads and reducing the read length to a common denominator (15 million and 50 bp, respectively). We have also used the standard ENCODE peak-calling pipeline.

To assess whether these steps were sufficient to make the results comparable and to examine the possible presence of major batch effects, we investigated in more detail which fraction of the 13,983 reproducible peaks shown in **Figure 2b** changed activity from week to week. This additional detail is now included as **Extended Data Figure 4**.

As expected, we observed that the fraction of peaks that changed activity from week to week (based on input data used to generate **Figure 2b**) is lower for adjacent weeks (e.g., weeks 4 to 5; or weeks 7 to 8; ranging from 28-39%) than for weeks set farther apart (e.g., week 4 to 8; ranging from 40-57%). This observation reinforces the general continuity of genome-wide enhancer landscapes across all stages examined. The fraction of enhancers with altered activity between two adjacent weeks increased continuously from 28% (weeks 4 to 5) to 33% (weeks 5 to 6) to 34% (weeks 6 to 7) to 39% (weeks 7 to 8), potentially reflecting an increasing complexity of cell types and morphogenetic processes during later weeks. Importantly, the difference between weeks 6 (containing only samples from Wilderman et.al., 2018) and week 7 (containing only new samples from our study) is at the lower end of the range provided by the adjacent week transitions (33%-39%), suggesting strong continuity between the two datasets.

While we cannot exclude the presence of minor batch effects due to various factors described above, we believe that these additional analyses support that such effects do not confound our general observations of temporal dynamics in major ways. We expect this analysis to be of significant value to the community, despite unavoidable imperfections when comparing data across studies. To make sure that the reader is aware of the limitations pointed out by the reviewer, we have now revised the main text to emphasize some of the aspects described here and added the following details:

pg 28, lines 605-606, Methods

For the ChIP-seq datasets at CS13-15, CS17 and 20²⁶, publicly available and post-mapped TagAlign files were used.

pg 29, lines 614-625, Methods

We note that the use of human embryonic samples, which are typically derived from individual, or a small number of fetal tissue donations can introduce variability regarding tissue dissection and genetic heterogeneity. While some of these sources of variation are unavoidable, we tried to minimize potential batch effects. To make the analysis as comparable as possible, we down-sampled the number of input reads and the read length to a common denominator (15 million and 50 bp, respectively), and used the standard ENCODE peak-calling pipeline. To assess the possible presence of batch effects between data from these studies, we compared temporal transitions between weeks (**Extended Data Figure 4**). In this analysis, we did not observe discontinuities specifically associated with the transition time points between batches. While we cannot exclude the presence of some batch effects, this result suggests that study-specific batch effects do not confound our temporal dynamics analysis in major ways.

3. Lines 182-186 and figure 2b: The numbers do not match. 1,624 continuously active enhancers + 3,137 continuously active enhancers in a subset of timepoints do not sum up to 5,373 as indicated in the figure. In figure 5b, it is not clear where are the 1,624 elements predicted to be continuously active as enhancers throughout all five weeks (line 182 – 184), and where are the 3,137 elements that showed continuous activity periods covering a subset of the five weeks (lines 185-186). Instead, we observe a block of 5,373 ‘continuous elements’, which does not correspond to the sum of 1,624 + 3,137.

Moreover, these data should be presented in a table, containing the coordinates of the active regulatory elements plus their week of activity.

We thank the reviewer for catching this inconsistency. The number reported in the text should have been 3,749 instead of 3,137. This has now been corrected. To avoid confusion, we now refer to enhancers that are active throughout all weeks as “constant”, to distinguish them from enhancers with “continuous” activity across several but not all stages examined. In total, the 13,983 elements include:

- 1,624 active throughout all five weeks (now termed “constant”)
- 6,374 active on one week only (“week-specific”)
- 2,236 that go from being active to inactive to active again (‘non-continuous’)
- 3,749 active throughout some portion of the five weeks (now termed “continuous”)

We have updated the **Figure 2b** accordingly. We have now also added this information to **Supplementary Table 2** to indicate clearly which category each of the 13,983 elements falls into. We also revised the main text as follows:

pg 10, lines 205-209, Results

We found that a small proportion (1,624 elements or 11.6%) of elements were predicted to be continuously active (labeled “constant” in **Figure 2b**) as enhancers throughout all five weeks. Nearly half (6,347) showed narrow predicted activity windows limited to a single week, while another 3,749 showed continuous activity periods covering a subset of the five weeks.

4. Line 143: how are defined the presumptive target genes of the human candidate craniofacial enhancers identified by the authors? The authors identified the nearest genes to specific enhancers as potential target genes for those enhancers. However, only a fraction of enhancers are known to target their nearest genes, as enhancers can show selective directed interactions with far away genes, often skipping proximal promoters.

This raises the possibility that other mechanisms, such as long-range chromatin interactions, may regulate the target gene-promoter interactions for the remaining enhancers.

Enhancers indeed show a wide range in activity based on the distances to their presumptive target genes. Broadly, while some reports suggest that few enhancers regulate their nearest genes (Sanyal et al., 2012 PMID: 22955621), others have shown that the nearest gene is the target gene for nearly half of all identified enhancers based on chromatin conformation data (Freire-Pritchett et al., 2017 PMID: 28332981). Large-scale genomic perturbations have shown that 47% of all enhancers target their nearest genes (Fulco et al., 2019 PMID: 31784727). Notably and of direct relevance to this manuscript, data from cranial neural crest cell differentiation show that species-biased (human vs chimp) regulatory regions are in fact enriched immediately next to genes important for craniofacial development (Prescott et al., 2015 PMID: 26365491). Nonetheless, we agree with the reviewer that in many cases the closest gene is not the target gene and have therefore performed additional analyses to provide improved gene-enhancer associations.

We have now intersected our list of 13,983 reproducible human enhancers with publicly available long-range chromatin interaction data for approximately 19,000 promoters from human embryonic stem cells (Jung et al., 2019 PMID: 31501517) and have included the most likely target gene for each enhancer in **Supplemental Table 7**. We have now added this analysis to our manuscript as below.

pg 9, lines 179-182, pg 10, lines 183-188, Results

In a complementary assessment, we explored the putative target genes of the human reproducible enhancers with predictions from publicly available promoter-centric long-range chromatin interaction data for approximately 19,000 human promoters⁴⁴. This interaction-based mapping strategy identified 3,005 chromatin segments containing predicted craniofacial enhancers interacting with the promoters of 2,921 nearby genes (**Supplemental Table 7; Methods**).

pg 31, lines 676-681, Methods
Enhancer-Target Gene Predictions

We intersected our list of 13,983 reproducible human enhancers with publicly available long-range chromatin interaction data derived from promoter capture HiC for approximately 19,000 promoters in human embryonic stem cells⁴⁴. Genomic coordinates of the interacting fragments were converted to hg38, the predicted target gene and extent of overlap with the human enhancers from this study are reported in **Supplemental Table 7**.

5. Lines 205-220. To assess whether the differences in epigenomic signatures between human and mouse translate into species-specific differences in *in vivo* enhancer activity, the authors used a transgenic mouse assay to compare the human and mouse orthologs of a predicted human-specific enhancer. They found that ‘embryos transgenic for the human ortholog (hs2656) show reproducible activity in the developing nasal and maxillary processes at embryonic day (e) 12.5, confirming that the human tissue-derived enhancer signature correctly predicts *in vivo* activity at the corresponding stage of mouse development (Figure 2c). In contrast, we did not observe reproducible craniofacial enhancer activity with the mouse orthologous sequence, concordant with the absence of enhancer chromatin marks in mouse at this location.’

It should however be mentioned that hs2656, while being conserved between human and mouse is not identical between these 2 species, thus probably explaining the differences in epigenomic signatures between mouse and human. What are the differences in the orthologs (human and mouse) that could potentially create differences in the *in vivo* assay when tested in the mouse system? If not in the result section, some explanation/ speculation in the discussion would be illuminating for the reader (see also below).

We thank the reviewer for this comment. Our underlying assumption was indeed that these differences in epigenomic signatures and *in vivo* activity are due to sequence differences between the two enhancer orthologs. However, we acknowledge that this was insufficiently explained in our manuscript. We have now added the following sentences in the Discussion to explain this better:

pg 22, lines 429-438, Discussion

We observed that enhancer hs2656, but not its mouse ortholog mm2280, shows craniofacial *in vivo* activity in transgenic mice. This is consistent with an epigenomic enhancer signature at this element in human, but not in mouse tissue. These lineage-specific differences in epigenomic signature and *in vivo* activity are likely due to sequence differences within the enhancer element itself, which may affect transcription factor binding sites or other functionally critical motifs embedded in the enhancer. For example, within the most conserved 425bp core sequence of enhancer hs2256, 31% of the nucleotide positions show differences between human and mouse, which include binding sites for transcription factors that are important for craniofacial development, such as TFAP2B and TCF4⁹²⁻⁹⁴.

6. Lines 131-134: 44 human enhancers could not be validated in the mouse despite their H3K27ac signature. Please comment on this in the light of the claim in lines 216-217 and figure 2c/d. From one example you cannot conclude that “human tissue-derived enhancer signature correctly predicts *in vivo* activity at the corresponding stage of mouse development” (lines 216-217). Which of these human enhancers are positive for H3K27ac after integration into the mouse genome? Additionally, the sequence conservation between hs2656 and mm2289 is not perfect and may explain differences in H3K27ac and functionality (see above). Therefore, the conceptual distinction between sequence and epigenetic conservation is elusive.

Epigenomic signatures are known to be an imperfect predictor of *in vivo* enhancer activity. This is the case even when signatures derived from mouse tissues are used in combination with a transgenic mouse reporter assay. We examined this in detail as part of the ENCODE project (Moore et al., 2020 PMID:32728249). Using a similar combination of ATAC-seq and H3K27ac marks from a panel of embryonic mouse tissues, we observed that tissue-specific validation rates range widely depending on tissue and intensity of epigenomic signal. While validation rates of 60%-75% were observed for the few candidate enhancers with the strongest genome-wide signals in each tissue, the majority of candidate sequences showed much more modest *in vivo* validation rates, e.g., ranging from 20% to 27% for candidate enhancers at rank-positions of ~3,000 in their respective data set (see Figure 4a in Moore et al., 2020). In addition, in this benchmarking study we observed substantial numbers of enhancers with *in vivo* activity in tissues other than those predicted by the epigenomic marks. Thus, the observed validation rate of 27% in the present study, as well as the observation of multiple enhancers with activity in non-craniofacial tissues, falls well within expectations for this type of enhancer prediction approach. For practical reasons it would be very challenging to assess the H3K27ac status of human enhancers following their genomic integration in transgenic mice, since this would require the generation of one or several stable transgenic lines for each enhancer examined, collection of embryonic tissues, and a separate ChIP-seq experiment for each enhancer examined. This would require major additional experimental effort and would represent a substantial expansion of the scope of our study.

Please see the previous point regarding the conceptual relationship between sequence, epigenomic, and functional conservation.

7. Lines 265 – 267 “we observed 16,564 differential accessible regions (DARs) across 20 separate clusters, indicating that each of the clusters representing identical or very similar cell types have distinct open chromatin signatures”. This sentence is not clear. How can the authors claim that the clusters identified by snATAC-seq represent identical or very similar cell types? Knowing that all tissues from the craniofacial prominences (i.e. epithelium, endothelium, mesoderm, neural crest cells...) were included in the analysis.

We apologize for this lack of clarity and potential overstatement of our conclusions. We have now modified this sentence as below.

pg 14, line 293, Results

...indicating that each of the clusters has distinct open chromatin signatures.

8. The authors claim that the putative enhancer regions near *Isl2* and *Scaper* (mm2285), and in the intron of *Lrrk1* (mm2282) drive reproducible lacZ-reporter activity in the developing mouse face in anatomical regions that are consistent with neuronal and epithelial cell types. This statement should however be confirmed by doing a double staining on sections (LacZ + a neuronal marker for mm2285 / LacZ + an epithelial marker for mm2282).

We agree that our data cannot support an explicit claim regarding the identity of the stained cell types. We have now revised this sentence to reflect this:

pg 15, lines 313-316, Results

These putative enhancer regions near *Isl2* and *Scaper*, and in the intron of *Lrrk1* drive reproducible lacZ-reporter activity in the developing mouse face in anatomical regions where neuronal and epithelial cell types are expected to be found (mm2285 and mm2282, **Figure 4c**).

Moreover, without knowing the target genes of these enhancers, it is hard to predict where the LacZ staining should be. The authors make the supposition that the target genes are the nearest ones, but it is not necessarily the case, it is even often wrong (see above). Along this line, we can appreciate that mm2285 does not reproduce the pattern of expression of *Isl2* or *Scaper*, and that mm2282 does not reproduce the pattern of expression of *Aldh1a3*.

Please see response to point #4 above regarding relationships between enhancers and the nearest vs more distal potential target genes.

In addition, we note that the number of enhancers in metazoan genomes far exceed the number of transcribed genes, and multiple enhancers are known to regulate a single gene, conversely a gene can be regulated by different sets of enhancers in distinct cell types (Kieffer-Kwon et al., 2013 PMID: 24360274, Panigrahi et al., 2021 PMID: 33858480). Therefore, it is not expected that the cell-type specific enhancer activity and the expression of its target gene are limited to a single cell type in time. In fact, the currently accepted paradigm in the field is that modularity, redundancy, and additivity of enhancer function drive tissue-specific phenotypes (Osterwalder et al., 2018 PMID: 29420474; Long et al., 2016 PMID: 27863239). This is consistent with what we see in our analyses in **Figures 4** and **6**.

We have now added the following details about the *in vivo* gene expression of *Isl2* and *Aldh1a3* based on previous reports from literature, which align with our observations of enhancer activities.

pg 15, lines 316-319, Results

Notably, the spatial expression pattern of mm2285 and mm2282 is consistent with the expression of *Isl2* in cranial ganglia^{88,89}, and the expression of *Aldh1a3* in the retina and the nasal epithelium⁹⁰ in similar developmental windows in mice *in vivo*.

The authors should then first determine the target gene of the enhancer, check that the LacZ reproduce the expression pattern or part of the expression pattern of this target gene, check in which cell type(s) this target gene is expressed, and check in their snATACseq data that their enhancer is indeed open in the cell type(s) where the LacZ is present.

We apologize for not clarifying this sufficiently in our description of the figure. In **Figure 4c**, we first picked candidate regions based on their accessibility signatures in the snATAC-seq data

(wiggle-tracks in **Figure 4c**, showing accessibility across all 20 clusters in our snATAC-seq data), and then tested these regions (mm2285 and mm2282) in our transgenic enhancer-lacZ reporter assay thus validating the predictions from *ScanFaceN*. Within the immediate vicinity of these two enhancer regions, we highlighted genes with positive expression in *ScanFaceX* and those that were reported in the OMIM catalog as human disease-causing and chose to show those in the main figure panel. We have now added the following sentence in the main text to clarify this for the reader.

pg 15, lines 305-307, Results

Within the immediate vicinity of these two enhancer regions, we display genes with positive expression in *ScanFaceX* and those that were reported in the OMIM catalog as human disease-causing.

9. Lines 316 to 331. To validate their cell type-resolved enhancer predictions, the authors intersected their data with craniofacial enhancers validated *in vivo* and curated in the VISTA Enhancer Browser. However, as mentioned above, it's not possible to determine the cell type specificity of the VISTA enhancers, only based on a LacZ staining performed on whole mount embryos. For instance, the authors claim 'the chondrocyte cluster (cluster 13) has multiple VISTA enhancers with activity in the mid-face, paranasal regions, and/or the region at the junction of the developing forebrain and nasal prominences that are consistent with developing cartilaginous regions of the face'. Looking at regions where an enhancer is active is not sufficient to make conclusions on the cell-type specificity of this enhancer.

We apologize for the lack of clarity. We fully agree with the reviewer and reiterate that we did not intend to make conclusions about the cell type based on whole mount images from VISTA. This part of the paragraph is now revised to the following:

pg 18, lines 358-366, Results

For example, the connective tissue-mesenchymal cluster (cluster 2) of the craniofacial snATAC-seq tends to group VISTA enhancers with activity specific to the branchial arches (**Figure 5b**). Despite broad correlations, we observed considerable heterogeneity of spatial patterns within most clusters. For example, the chondrocyte cluster (cluster 13) has multiple VISTA enhancers with activity in the mid-face, paranasal regions, and/or a region at the junction of the developing forebrain and nasal prominences that may constitute the developing cartilaginous regions of the face (**Figure 5b**). These observations underscore the spatiotemporal complexity of craniofacial morphogenesis, which relies on cell type-specific regulatory programs in combination with highly regionalized regulatory cues.

The last part of the analysis performed by the authors, called "Cell Type-specific Enhancer Activity at Single-cell Resolution" (from line 340) indeed explores the cell type specificity of selected craniofacial enhancers *in vivo*, but this analysis is performed on only 3 enhancers and none of them are specifically active in one cell type (presence of multiple red points for each of the

enhancer tested, figure 6c). Of note hs746 is not only active in osteoblasts as claimed in the text, but also in the eye (as seen on the figure 6a and 6b). Similarly, hs521 is not only active in a subset of mesenchymal cells and chondrocytes progenitors, as claimed in the text, but also in the eye (as seen on the figure 6a and 6b).

Previous studies have established that an enhancer can be active in one or multiple cell types (specificity can be obtained by lineage-specific transcription factor binding or other cues), or can even have ubiquitous activity (Andersson et al., 2014 PMID: 24670763). In fact, the number of enhancers in the metazoan genome far outnumbers the number of transcribed genes, and multiple enhancers are known to regulate a single gene, conversely a gene can be regulated by different sets of enhancers in distinct cell types (Kieffer-Kwon et al., 2013 PMID: 24360274, Panigrahi et al., 2021 PMID: 33858480). For example, of the 1,681 positively validated *in vivo* enhancers that have been characterized in the VISTA enhancer browser to date, 856 (51%) have reproducible activity in more than one annotated embryonic structure. While the VISTA enhancer browser is based on whole-mount data and does not annotate cell types, the proportion of enhancers with activity in more than one cell type is likely even higher.

Assuming that the enhancers examined in **Figure 6** are one of several enhancers of their respective target genes, it is expected that the enhancers (panel **a**) may show activity in only a subset of the cells in which the target gene is expressed (panel **c**, bottom). This is consistent with what we see in our analyses in **Figure 6**.

We agree that the cleanest examples of this could be provided by cases in which both the enhancer and the expression of its target gene is strictly limited to a single cell type within a single embryonic tissue and subregion thereof, and without any other nearby enhancers that may have overlapping activity patterns. However, we found it challenging to identify such an ideal example within the enhancers (and presumptive target genes) that we validated *in vivo* in the course of our studies.

10. Line 99: The RNAseq data introduced here is not mentioned anywhere later in the manuscript. It is also listed in supplementary table 1 though. Please specify why it was included in the manuscript and report the results. The RNAseq data could be used to correlate expression and H3K27ac levels across stages, for example. This would be particularly interesting for the predicted stage-specific enhancers. Please check and report whether the corresponding genes are also generally expressed at lower levels in that specific stage.

We agree that the available RNA-seq data included in our manuscript was minimally described and we thank the reviewer for pointing out this missed opportunity. We have now used this data, as suggested by the reviewer, to examine correlations between expression and H3K27ac levels at enhancers.

As described in the response to this reviewer's major point #4 above, we leveraged the intersections of our list of 13,983 reproducible human enhancers with publicly available long-range chromatin interaction data for approximately 19,000 promoters from human embryonic

NCOMMS-23-28972: Cell Type- and Tissue-specific Enhancers in Craniofacial Development by *Rajderkar et al.*,

stem cells (Jung et al., 2019) and the most likely target gene for each enhancer thus assigned (**Supplemental Table 7**). For 3,005 predicted craniofacial enhancers (this study) overlapping with chromatin segments from Jung et al., 2019 and interacting with the promoters of 2,921 target genes, we assessed the correlation between TMM-normalized ChIP-seq signal for H3K27ac-bound regions of interest with gene expression (RNA-seq) counts for which both ChIP-seq as well as RNA-seq data were available from the exact same biological samples (matched samples: CS18_12612, CS18_12695, CS19_12696, CS22_11963, and CS23_12492). We have now included the results of this analysis in the manuscript as below.

pg 9, lines 179-182, pg 10, lines 183-188, Results

In a complementary assessment, we explored the putative target genes of the human reproducible enhancers with predictions from publicly available promoter-centric long-range chromatin interaction data for approximately 19,000 human promoters⁴⁴. This interaction-based mapping strategy identified 3,005 chromatin segments containing predicted craniofacial enhancers interacting with the promoters of 2,921 nearby genes (**Supplemental Table 7; Methods**). Across 2,263 predicted gene-enhancer pairs with epigenomic enhancer predictions and gene expression data available from identical biological samples, we observed a positive correlation between sample-specific enhancer activity and gene expression levels ($p=0.00002$; Mann-Whitney U Test; see **Extended Data Figure 3, Supplemental Table 8; Methods**).

pg 31, lines 681-693, pg 32, lines 694-702, Methods

For 3,005 chromatin segments containing predicted human craniofacial enhancers, and interacting with the promoters of 2,921 genes, we performed Spearman's Ranked Correlation Coefficient (SRCC) analysis between enhancer signal intensities (H3K27ac ChIP-seq, Trimmed Mean of M-values normalized) and gene expression counts (RNA-seq) of the assigned target genes (**Supplemental Table 7**) for predicted enhancer:target gene pairs versus all other pairs. We performed this analysis for combined as well as individual activity windows shown in **Figure 2b** for a subset of matched samples, i.e., five instances where enhancer predictions and gene expression data were available from identical human embryonic face samples, namely CS18_12612, CS18_12695, CS19_12696, CS22_11963, and CS23_12492 (**Extended Data Figure 3, Supplemental Table 8**). Mann-Whitney U test statistic was used to ascertain significance between the correlated enhancer:target gene pairs of interest versus all other pairs.

We note that the correlation is highly significant but quantitatively moderate. This is likely due to technical factors including imperfect enhancer-gene associations, target gene predictions not being available for all enhancers, differences arising from comparing predictions from human embryonic stem cells versus complex primary human embryonic tissue encompassing varying stages of differentiation, not excluding cases with redundant enhancers acting on the same gene(s), and uncertainty about the expected quantitative correlation between H3K27ac signal intensity at an enhancer and the expression level of a target gene. For the correlation for class "week-specific" in **Extended Data Figure 3b**, the comparisons may not be significant due to the lack of capability of SRCC to detect patterns driven by one or two data points.

11.a) Lines 101-102: Please provide more details on how the data from Wilderman et al. was integrated into your analyses. For example, which regions from Wilderman et al. were used (ChromHMM predicted enhancers or H3K27ac peaks)?

Please see response to point 2 above.

11.b) Similarly, please also specify what was considered “an epigenomic enhancer signature in the mouse” (lines 199-200) which is currently unclear.

We apologize for the incorrect phrasing; it is now modified as below.

pg 11, lines 226-228, Results

Among these conserved sequences, 8,257 (59%) of the human candidate enhancers showed H3K27ac binding in the mouse, indicating their functional conservation.

11.c) Additionally, in line 105 the number of enhancers in later stages (CS18-23) is mentioned (10,893 regions, data generated in this study). It is left unclear whether there are only ~3,000 enhancers found in earlier stages (Wilderman et al. data) due to generally lower numbers of enhancers identified by Wilderman et al.

3,090 enhancers are exclusively active primarily before CS18, with one sample at CS20 from Wilderman et al., 2018. Also see our response to point #1.

11.d) Roughly, the total number of enhancers per week seems different (figure 2b) with less enhancers found in the stages/weeks only covered in the Wilderman et al. dataset. Technical differences are likely reinforced by varying number of replicates for different weeks. Consequently, the limitations mentioned in lines 188-189 apply to all groups of enhancers (week-specific, continuous, non-continuous). Therefore, please adapt and expand this statement on the limitations with respect to the temporal dynamics of enhancer activities.

Thank you for pointing this out. While there is not a strong correlation between the week of embryonic development, the number of samples per week and number of peaks per week (e.g., week 4 has 5272 peaks, and week 5 has 2916, despite having 5 and 6 samples, respectively), all these variables, as well as differences in quality between individual samples, could influence the results to some degree. While we do not have sufficient numbers of samples to estimate the effect of each of these variables, we have now revised the main text to make the reader more aware of these limitations:

pg 11, lines 213-216, Results

We note that the analysis of temporal dynamics of subsets of enhancers may potentially be influenced by the variable number of samples or peaks per week. However, we do not

observe obvious confounding effects due to these variables within the samples we have analyzed (**Extended Data Figure 4, Supplemental Table 11; Methods**).

pg 29, line 613, Methods

For a break-down of samples as well as peaks per week, see **Supplemental Table 8**.

12. Line 497: Did the peaks have to be present in at least 2 replicates of each week? This would mean that no week-specific peaks can be identified, which is in contrast with the results of figure 2b. In case they were only required to be present in at least two replicates of one, i. e. the same, week, the term “highly reproducible” (l. 543, l. 152) seems to be an overstatement. Please clarify.

We have indeed required only that a peak is present in two independent samples (different embryos) from the same week (as described both in the main text and in **Methods**), after filtering for FDR<5%. We have now removed references to “highly reproducible” sets of peaks instead calling them simply “reproducible”.

pg 9, lines 165, Figure 2, Legend

Results of rGREAT ontology analysis for 13,983 reproducible human craniofacial enhancers..

pg 32, lines 712-714, Methods

...was intersected with 13,983 reproducible human enhancers derived from primary embryonic bulk face between CS13-23.

13. Line 125: How much overlap of the VISTA enhancers and the enhancers identified in this study set would you expect to see by chance? This is considering general mappability, accessibility, etc. of the genome.

We thank the reviewer for this suggestion, which often comes up for these types of studies. However, this analysis cannot be done in a meaningful way because VISTA enhancers are not a random sample of the genome. Most were intentionally picked for their high levels of evolutionary conservation, high levels of epigenomic signal in embryos, lower repeat content, and/or proximity to genes known to regulate embryonic development. Many were selected based on a variety of tissue-specific epigenomic signatures as part of previous targeted validation efforts, which included craniofacial epigenomic data sets. In many cases, candidate sequences in VISTA were tested based on multiple supporting lines of evidence, making it impossible to disentangle and normalize for different predictors retrospectively. Thus, the VISTA dataset cannot be used as a valid neutral proxy to quantify enhancer enrichment compared to genome-wide background rates. We have now added the following note in the main draft to explain this:

pg 33, lines 726-729, Methods

We note that VISTA enhancers are not a random sample of the genome and are intentionally picked for their high levels of evolutionary conservation, high levels of

epigenomic signal in embryos, lower repeat content, and proximity to genes known to regulate embryonic development.

14. Line 132: How were the 60 enhancer candidates selected? Please also add a statement to lines 133-134 about how many of the 60 tested enhancers showed activity in other structures than the face.

We have now included the following details on the selection criteria for these 60 enhancer candidates in **Methods**.

pg 27, lines 556-562, Methods

The 60 candidate enhancer elements were selected based on a combination of criteria including overlap with ATAC-seq peaks, strength of H3K27ac active enhancer signatures, non-mouse annotated regions, and vicinity of genes with known or proposed roles in craniofacial development based on human genetics and/or mouse knockout studies (e.g., genes listed under term “abnormality of the face”; HP:0000271 in Human Phenotype Ontology¹¹⁰ or “craniofacial abnormalities”; MP:0000428 in the Mammalian Phenotype Browser¹¹¹).

We have now added the following details on the activity of the 60 tested enhancers in the main draft.

pg 8, lines 144-146, Results

Of these, a total of 28 candidate enhancers were positive for reporter activity out of which we identified 16 cases of previously unknown enhancers that showed reproducible activity in craniofacial structures.

pg 8, lines 151-154, Results

Of the 16 enhancers positive for craniofacial tissues, 8 were simultaneously active in non-craniofacial structures such as the brain or limb, while the remaining 12 out of the total 28 were only positive in non-craniofacial tissues (**Supplemental Table 5**).

15. Line 144: The rGREAT analysis is missing from the methods section. How were target genes assigned to enhancers? For how many enhancers could you identify a target gene? Against which background was ontology enrichment performed?

We apologize for missing this description in the manuscript. We have now added the following in **Methods**.

pg 31, lines 666-675, Methods

rGREAT Ontology Analyses

To identify human phenotype ontology terms enriched in our list of 13,983 reproducible human craniofacial enhancers, we ran rGREAT⁴³ (Bioconductor version: Release 3.17) that performs GREAT¹¹⁷ analysis (<http://great.stanford.edu>) on non-coding regions to

predict their functions based on annotations of nearby genes. Following parameters were used from the GREAT tool: a default of 5kb upstream and 1kb downstream basal plus extension for proximal regulatory regions, up to 10 kb for distal regions, and curated regulatory domains were included. A background of whole genome hg38, a cut-off based on Binomial False Discovery Rate < 0.01, and Fold Enrichment > 2 was applied to retain the top terms (**Supplemental Table 6**).

For enhancer-target gene identification, please see major point 4 above.

16. Lines 171-177: It is unclear how this analysis supports the identified enhancers, since only LD SNPs that are inherited together with the putative causal SNPs overlap with enhancer candidates and no causal SNPs itself.

We thank the reviewer for highlighting this general challenge in the interpretation with GWAS data. It is now well established that the lead SNPs identified by GWAS (i.e., those with the strongest statistical signal in a given population) are often not the causal SNPs, due a variety of complex statistical and population structure effects (see Schaid et al., 2018 PMID: 29844615 for discussion). We therefore included all SNPs in strong LD with the lead SNPs in our analysis.

Line 175: Is there an enrichment of LD SNPs in the enhancers or is it just an overlap you may see by chance?

We have now refined our enrichment analysis. We observe that normal face variation and human craniofacial disease LD SNPs are highly significantly ($p < 10^{-8}$) enriched in variation- or disease-associated lead and LD SNPs, compared to a genome-wide set of control SNPs. Quantitatively, the fold-enrichment is moderate (odd ratio = 1.27), which is expected because the majority of combined lead and LD SNPs is to be non-causal. We also did not attempt to exclude the large number of SNPs in moderate LD with lead SNPs from the control set, which might have further increased the enrichment. In summary, these findings support our initial conclusion that enhancers identified in our study are enriched in genetic signals associated with normal facial variation or craniofacial disease. We have now included the following detail about this enrichment analysis as below.

pg 10, lines 197-198, Results

This LD SNP density represents an enrichment compared to control SNPs not implicated in craniofacial traits (OR = 1.27, $p < 10^{-8}$; **Methods**).

pg 32, lines 712-721, Methods

This compilation of GWAS (**Supplemental Tables 9-10**) was intersected with 13,983 reproducible human enhancers derived from primary embryonic bulk face between CS13-23. We have partitioned a total of 14,137,504 SNPs from the dbSNP155^{120,121} catalog by their association with normal face variation or human disease and overlap with reproducible fetal human face enhancers described in this work. We found that 605 out of

27386 (2.3%) of normal face variation- or human disease-associated SNPs overlapped the peaks, while only 245,727 out of 14,083,942 (1.8%) of non-associated SNPs did. The overlap was significantly different from random expectation with an odds ratio of 1.27 (Pearson's Chi-squared test with Yates' continuity correction: $X^2 = 34.102$, $df = 1$, $p\text{-value} = 5.229e-09$).

17. Fig 3 and 4: Why did the authors analyze different stages of embryonic development for scRNAseq and snATACseq? Do the conclusions still hold for the subset of snATACseq from only E11.5-13.5 (corresponding to the stages used in scRNAseq)? Please include results from this analysis of a subset of the data. Why are replicate numbers from the stages different?

We attempted to collect matched timepoints and for different assays wherever possible. However, not all samples passed our final quality control criteria. In the interest of considering the deepest data set possible for analysis, we included all data sets that passed quality control in our final analysis, rather than excluding individual samples because a matched assay did not yield high-quality data for the same developmental stage. The retained stages did not affect the distribution of relevant cell types in either data type (see **Extended Data Figures 8 and 14**) and are therefore unlikely to have skewed the overall analysis in significant ways.

18. Figure 4b: The authors picked the TSS with highest correlation between ATACseq and RNAseq data to investigate correlation between the data modalities, which will bias the analysis. Please consider picking the TSS with the highest accessibility instead.

Frequently, the most accessible TSS corresponds to a housekeeping gene, exhibiting accessibility across all cell types but lacking the variability between cell types necessary to establish a correlation between TSS accessibility and expression. While TSS accessibility and expression often display a correlation due to the dependence of expression on promoter accessibility, the influence of various factors on a gene's steady-state transcript levels can result in a tentative correlation (de la Torre-Ubieta et al., 2018 PMID: 29307494, and Starks et al., 2019 PMID: 30795793). A more suitable approach involves evaluating the correlation between TSS accessibility and gene expression of established cell type marker genes, as we demonstrate in **Figure 4b**. We chose to display these correlations for *Epcam*, *Dsp*, *Cthrc1* and *Cldn5* to demonstrate examples for select marker genes of general interest to craniofacial biology.

19. Figure 4b: What does the width of the bars indicate? And why do they have the same height?

We apologize for the confusing visual representation using bar-like visual markers. Neither the width nor the height of the individual bars contained any information (they were all the same, except for their color). We have now rectified this by replacing the bars with round dots, there are no changes to the underlying analysis.

20. In figures 1b, 2c, 2d phyloP scores are shown, while placental conservation is depicted in figure 4c. Please describe which phyloP score was used and why you switched to placental conservation in figure 4c.

We apologize for the mis-label in **Figure 4c**. It is now corrected to PhyloP.

21. Line 270: The authors describe the number of DARs retained, but not the number of clusters out of the 16 annotated cell types from ScanFaceX that could be retained or the percentage of single cells from the ATACseq data that could be assigned to a cell type.

We apologize for missing this detail in the description, the sentence is now modified to reflect that 11 clusters out of 16 annotated cell types from *ScanFaceX* were retained in the integrated snATAC-seq data.

pg 14, lines 296-297, pg 15, lines 298-299, Results

Upon integration, a substantial subset of DARs (10,038 out of 16,564; 60%) were retained, and 11 annotated clusters from developing craniofacial cell types including chondrocytes, myocytes and connective tissue, epithelial cells, and sensory neurons showed high correlation between the two data types.

22. Line 322: Are these 77 VISTA confirmed snATACseq enhancers already among the previously identified 153 H3K27ac-enhancers (line 125) or different ones?

Yes, these 77 are from previously validated VISTA enhancers. We have now revised the text to clarify.

pg 18, lines 356-357, Results

...among 77 previously validated VISTA enhancers that showed chromatin accessibility..

It may simplify the readers' understanding to include a summary figure or table for numbers of enhancers and corresponding overlaps from: Wilderman et al., H3K27ac enhancers from this study, VISTA, snATAC from this study after liftOver

We have now added a summary table describing the various overlaps and intersections of the 13,983 human enhancers identified in this study with other experimental modalities in the present study as well as other publicly available craniofacial datasets.

Please see **Supplemental Table 21**.

23. Please expand the discussion section. Add comments on limitations (identification of enhancers through other methods and validation, assignment of target genes, integration into H11 locus which takes the genes and enhancers out of their 3D context, etc.). Discuss transferability between humans and mice, too.

We have now added the following points in the Discussion.

pg 23, lines 458-464, Discussion

In this study, we leveraged genome-wide profiling of H3K27ac binding for identification of enhancers. The tissue-specific validation rate we observe is comparable to that we observed in other studies using similar methods for prediction of *in vivo* enhancer activities²⁵. We note that alternative experimental approaches that measure non-coding RNAs or massively-parallel reporter assays with or without mutational screens can also be used for identifying putative enhancer elements and may be useful for capturing additional craniofacial candidate enhancers^{99,100}.

pg 24, lines 483-488, Discussion

One of the limitations of present methods is the ability to capture low-expressing genes or rarer cell populations among other technical and statistical challenges^{102,103}. We note that utilizing cell type annotations from *ScanFaceX* and integrating those with single-cell open chromatin data provides correlative but not definitive evidence for the target genes of a given enhancer, which requires verification through complementary experimental methods¹⁰⁴⁻¹⁰⁶.

pg 24, lines 493-498, Discussion

We note that *in vivo* transgenic reporter assays can demonstrate that an enhancer is sufficient to drive expression in a tissue or cell type of interest, but integration into a safe harbor locus such as H11 removes the enhancer from the full epigenomic and three-dimensional context of its native locus¹⁰⁷. Therefore, reporter expression may not fully recapitulate the full endogenous activity of a given enhancer in its original genomic location.

Additional methodological points:

1. QC Steps: While it is implied that quality control (QC) steps were performed during the analysis, it would be beneficial to explicitly mention the QC criteria used and any additional steps taken to ensure data quality and reliability.

Thank you for pointing this out. We have now included the following additional details describing the QC steps in Methods.

pg 34, lines 753-756, Methods

Based on the inspection of UMI/gene count plots, the UMI range which preserved the main group of cells and excluded both droplet debris and likely clumps of cells was established for each sample separately (2,000-4,000 minimum, 15,000-60,000 maximum).

2. Reproducibility: It's essential to mention the versions of software/tools used in the analysis, as well as any specific parameters or configurations applied.

We apologize we missed listing the software versions in the following instance.
The details are now revised:

pg 36, line 821, Methods

..snapATAC2 (version 1) package

3. Adding Specific Metrics: Some key quality metrics (e.g., the number of cells, median genes per cell, mitochondrial percentage, etc.) after each preprocessing step could provide a better understanding of the data's quality and how filtering was applied.

Please see response to Additional methodological points, point#1 above.
In addition, we have added the following details describing key metrics for the *ScanFaceX* data, with an additional plot in the supplementary information.

pg 34, lines 763-766, Results:

Our final Seurat/clustered UMAP consists of a 25,645 feature by 57,598 cell matrix, with a median of 1,659 and a range of 500 - 8,840 genes expressed per cell (**Extended Data Figure 5**), and a range of 474-9,148 cells for the smallest to largest clusters (**Supplemental Table 16**).

4. Batch Correction Method: It is mentioned that batch effects were corrected, but the specific method used for batch correction should be stated explicitly.

We have now added the following details to describe how we processed each of the *ScanFaceX* samples and integrated them for downstream analysis.

pg 34, lines 756-761, Methods

For scRNA-seq, samples were integrated using standard Seurat procedures; *SelectIntegrationFeatures* function was run on a list of all 9 samples to be integrated to find 3000 most variable features. *mCherry* transcripts, genes on chromosomes X or Y (Gencode vM24) and cells expressing >5% mitochondrial genes (with names starting with mt) were removed from that list. *PrepSCTIntegration*, *FindIntegrationAnchors* and *IntegrateData* functions were run to obtain an integrated dataset.

5. Visualization of QC Results: Including a few plots or summary statistics related to the QC steps (e.g., gene count distributions, mitochondrial percentage distributions). The analysis and methods described for clustering and cell-type annotation using the snapATAC package appear well thought out and comprehensive. However, there are a few suggestions for clarity and potential improvements:

a) Method Parameters: For reproducibility, authors should consider providing specific parameter settings used for the snapATAC package and the Leiden algorithm for clustering. Mentioning the versions of software/tools used is also helpful.

We used the default settings and parameters unless stated otherwise. We have now specifically included the following details as requested.

pg 37, lines 838-839, Methods

..... (leiden clustering resolution =1) to define 20 clusters.

pg 38, lines 865-867, Statistical Analyses (note added)

Unless otherwise stated, default parameter settings were employed for any software tool that was used in the analyses.

b) Explain Data Transformation: While it is mentioned that the read coverage of bins was normalized with $\log_{10}(\text{count}+1)$ and Z-score transformation, a brief explanation of why these transformations were chosen and how they benefit the analysis could be included.

We have now added the following details to Methods.

pg 36, lines 828-830, Methods

To stabilize the variance and reduce the impact of noise, we normalized the read coverage of all bins with $\log_{10}(\text{count}+1)$, applied Z-score transformation to ensure that each feature contributes equally to downstream analyses...

c) Data Subset Choice: Explain the rationale behind choosing 30,000 "landmark" nuclei for the dimensional reduction step and clarify how this subset was selected to ensure it is representative of the overall dataset.

We mentioned previously that the memory usage of the matrices scales quadratically with the number of nuclei (alluding to demand on computational power). The rationale behind using 30,000 landmark nuclei is now explicitly elaborated as below.

pg 37, lines 834-835, Methods

Therefore, given the computational limitations at the time of analysis, and based on evidence provided by SnapATAC¹²⁸, we sampled a subset of 30,000 "landmark" nuclei...

d) Annotation Transfer: Describe the rationale behind using the default parameters for the FindTransferAnchors function in Seurat. If there were any manual adjustments made, it would be helpful to mention them.

Rationale for using *FindTransferFunctions* is now appended in the description.

pg 37, lines 845-847, Methods

The *FindTransferAnchors* function in Seurat uses unsupervised identification of anchors representing cells from separate datasets, with the assumption that these cells are derived from shared biological states¹²⁹.

No manual adjustments were made to the *FindTransferAnchors* function in Seurat.

Minor points:

We thank the reviewer for their detailed reading and have addressed the corrections stated in #1-9 below.

1) Figure legend of figure 1: The abbreviation PA2 is not explained. Please add pa2 to the list of abbreviations in the figure legend.

Rectified, **Figure 1, Legend, pg 7, line 131.**

2) Graphical abstract, figure 1a, figure 2c: Depiction of human and mouse - why is the yellow colour used for the human figure? We also suggest removing the background colors in figures 2c and 2d.

The human figure cartoon in **Figure 1a** is now shown in black, **pg 7**.
Background colors in **Figure 2c-d** are removed, **pg 9**.

3) Extended data figure 1: What do the asterisks and the purple color stand for? Please explain in the figure legend.

The legend for the said figure, numbered **Extended Data Figure 2** in the current manuscript version now explains the asterisks and purple color usage in the figure with more details.

pg 40, lines 933-935, Extended Data Figure 2, Legend

Grouped terms “any marked*” and “any 6 craniofacial” are shown in purple. Terms for facial mesenchyme, branchial arch and nose are each marked by an asterisk and constitute “any marked*”, while “any 6 craniofacial” comprises the six craniofacial terms shown here.

4) Figure legends of figures 4b and 4c: We suggest adding information that colours in figures 4b and 4c correspond to the colour code used in figure 4a.

Corrected, **pg 17, lines 344-345, Figure 4b-c, Legend.**

Colors in **(4b)** and the individual snATAC-seq tracks in **(4c)** correspond to the color code used in **(4a)**.

5) Figure 4c: The LacZ staining for the intronic enhancer is barely visible. You may indicate it with arrows, for example.

Corrected, **pg 17, Figure 4c.**

Added arrowheads to guide the reader.

6) Suppl. table 1, row 6: Sample name (column 1) includes CS19, while stage (column 4) says CS18. This is contradictory. Please correct.

Corrected.

7) Suppl. table 2: Please add an additional column to indicate in which weeks and/or Carnegie stages the respective H3K27ac peak was seen.

Added an additional column to indicate which week each of the H3K27ac peaks was found, **Supplemental Table 2.**

8) Suppl. table 13: The header of column 7 contains “annoated”. Please correct typo.

Corrected.

9) Suppl. table 1: The header of the table contains “detaisl”. Please correct typo.

Corrected.

10) Suppl. table 5: There is one single entry with stage e12.5. Is this correct or was this maybe unintentionally added e. g. by Excel’s autocompletion?

This is correct. No change needed.

11) Line 35: This seems to be an overstatement, since a fraction of the enhancers was already identified before by Wilderman et al. Please modify.

We apologize for any potential overstatement. Please see response to this reviewer’s major point#11c above.

We have now clarified that the ~14,000 human enhancers include ~5,000 that were newly identified in our study, and ~9000 for which our study provides an extended temporal activity profile across human prenatal development.

We have now modified the relevant text as follows:

pg 3, lines 36-40, Abstract

In total, we provide detailed temporal activity profiles for 14,000 human craniofacial enhancers across nine developmental stages from weeks 4 through 8 of human embryonic face development, which includes 5,000 newly discovered candidate
NCOMMS-23-28972: Cell Type- and Tissue-specific Enhancers in Craniofacial Development by *Rajderkar et al.*,

enhancers as well as refined temporal profiles for 9,000 previously described candidate enhancers.

12) Line 43, line. 200: Functional conservation is claimed based on the presence of histone marks, which do not prove functionality.

Please see response to this reviewer's major comment #11b above.
In addition, we have now modified the relevant sentence in the Abstract as below.

pg 3, lines 46-48, Abstract

.....we find that the majority (56%) of human craniofacial enhancers share accessible chromatin signatures in the mouse, and provide cell type- and embryonic stage-resolved predictions of their *in vivo* activity profiles.

In addition, please see response to Reviewer 3, major point#3.

13) Line 197: How did you define "alignable orthologous sequence"? Please specify and provide more details on this.

We apologize for the lack of clarity here. We have now rephrased the relevant sentence to reflect the exact tool we used to look at conserved sequences.

pg 11, lines 224-225, Results

...presence of alignable sequence using LiftOver (UCSC Genome Browser⁴⁸)...

14) Line 199: Please indicate that the percentage is in relation to the total number of human enhancer candidates (and not to the mouse enhancers).

Corrected.

pg 11, lines 226-228, Results

Among these conserved sequences, 8,257 (59%) of the human candidate enhancers showed H3K27ac binding in the mouse, indicating their functional conservation.

15) Line 266: Please remove "representing identical or very similar cell types". Up to this point in the text, you haven't shown yet how the snATACseq clusters translate to cell types.

Removed said phrase and replaced as below.

pg 14, line 293, Results

...indicating that each of the clusters has distinct open chromatin signatures..

16) Line 492, line 520: Why did you use hg19 and hg38 for different analyses?

We apologize for the confusion. In fact, line 492 was correct, all human datasets were aligned to hg38. The statement in line 530 (referring to hg19) was wrong. We have rectified this in the **Methods** describing RNA-seq, **pg 30, line 662**.

17) Line 494: Please provide more details on identification of overlapping peaks. `overlap_peaks.py` seems to take only 2 input files at a time. How did you handle this with your replicates? Please expand methods section accordingly.

We have now added more details as below.

pg 28, lines 606-609, pg29, line 610, Methods

Peak calling was performed using MACS2 (v2.2.4; --broad flag, q-value < 0.05); upon broad peak calling and applying the FDR filter, bed files were combined and merged using `bedtools`¹¹⁴. A combined peak set was called by merging peaks from all samples, and overlapping peaks for each sample were counted using `overlap_peaks.py`.

18) Line 511: Have you performed all steps of the ENCODE ATACseq pipeline, including IDR, pseudoreplicated peaks, etc? Please provide details.

We have performed the ENCODE ATAC-seq analysis step-wise and have now added the following details in the manuscript.

pg 29, line 637, pg 30, lines 638-643, Methods

ATAC-seq data was analyzed using the ENCODE ATAC-seq (unreplicated) pipeline (<https://www.encodeproject.org/pipelines/ENCPL344QWT/>). Briefly, reads were aligned the Bowtie2 aligner and filtered to remove unmapped and non-primary alignments, low quality reads as well as PCR duplicates. A subsample of 15 million reads was used as input to peak-calling, adjusted for Tn5 shift reads and sets of biological samples were assembled along with pseudoreplicates. Peak calls excluded ENCODE blacklist regions¹¹⁵ and peaks were assessed at an Irreproducible Discovery Rate of 0.05.

19) As an additional validation for the identified enhancers, could you please check and report the overlap with the FANTOM5 enhancer atlas, specifically from the HEPM cell line?

We thank the reviewer for this useful suggestion, overlaps with FANTOM5 - HEPM cell line are now included in **Supplemental Table 2**, and in a summary in **Supplemental Table 21**.

20) Lines 481-482: The word "with" is used twice next to each other. Please remove one occurrence.

Corrected.

21) Lines 124 to 129, instead of indicating the total number of regulatory elements that have been tested to date in both human and mouse in VISTA (3193 mentioned in the paper, 3320 when this reviewer checked on the site), it would be more appropriate to mention only the craniofacial regulatory elements, and only in Human.

If in VISTA I select the same criteria as the authors did in the supplementary figure 1 to select for the craniofacial enhancers (i.e. ear, branchial arch, eye, melanocytes, nose, cranial nerve, facial mesenchyme, mesenchyme derived from neural crest, trigeminal V (ganglion, cranial)) and in human, I found 254 elements. It means that among these 254 human craniofacial regulatory elements, they have identified 153 cases with their present human-derived epigenomic dataset.

Prompted by the reviewer, we revisited and refined this analysis. We focused on a core set of tissue-specific VISTA enhancers annotated for branchial arch, facial mesenchyme, and nose, since these structures are expected to make up a significant portion of the cells present in the samples. We observed that out of 130 human VISTA enhancers with reproducible expression in one of these three structures, 38 (29%) overlapped one of the 13,398 reproducible human enhancers predicted in our study. We also considered the possibility that our reproducibility requirements underestimated the extent of overlap between predicted and in vivo validated enhancers. Indeed, the overlap increases to 70/130 (54%) if we consider the full set of 70,000 peaks present in at least one of our human samples. We have now updated this in the manuscript as follows:

pg 7, line 136, pg 8, lines 137-139, Results

Among the 130 human craniofacial regulatory elements that have been tested in VISTA to date and that are annotated for branchial arch, facial mesenchyme or nose, we identified 38 cases (29%) with overlaps with an enhancer predicted through the present human-derived epigenomic dataset (**Extended Data Figure 2, Supplemental Table 4**).

pg 33, lines 723-725, Methods

We intersected a subset of 130 human craniofacial regulatory elements (out of 3,193 total curated) in the VISTA Enhancer Browser with 13,983 reproducible human candidate enhancers for weeks 4-8 from this study...

pg 40

Extended Data Figure 2 is now replaced and revised to show these changes.

22) Supplementary figure 4 and extended data figure 1: it is not specified from which species(s) the 506 craniofacial enhancers were selected in the VISTA enhancer browser (mouse? human? both?). It seems that they were selected from both human and mouse.

As the authors' aim was to compare the human craniofacial enhancers they identified with the enhancer's reporter in the VISTA enhancer browser, it would be more appropriate to select only the human enhancers in VISTA.

Please see response to point #21 above.

23) Supplementary table 5: the meaning of abbreviations is missing.

Meaning of abbreviations is now added in **Supplemental Table 5**.

24) Extended data figure 3: the gene names (left side) are not visible.

We apologize for the gene names not being clear, we have now increased the font size of the gene names in the now revised **Extended Data Figure 7, pg 45**.

25) Extended data figure 5: (top) the names of the cell types are not visible.

We apologize for not noticing this at the time of submission, the names of cell types in the top half of the figure are now written in a bigger font for this figure which is **Extended Data Figure 9** in the current version, **pg 47**.

Reviewer #2 (Remarks to the Author)

This study performs an impressive amount of bulk RNA, ATAC, and H3K27Ac sequencing of human embryonic craniofacial tissues at various stages, as well as single-cell RNA and ATAC sequencing of mouse embryonic craniofacial tissues. It clearly provides an important and extensive resource for researchers interested in the gene regulatory basis of human craniofacial variation and disease. Another strength is the extensive validation of human enhancers in mouse, with 1/4 of those tested showing craniofacial expression. A large amount of new and published datasets, including extensive GWAS data, are integrated, and all datasets are publicly available to the community. Before publication, some issues with data interpretation should be addressed. The cell type annotations in places appear inaccurate, such as the cluster referred to as osteoblasts. This may reflect reliance on automated cell cluster calling rather than manual annotation based on the large number of single-cell studies that have already been performed of the craniofacial complex. A general weakness of the paper is the lack of citations of single-cell studies published by other groups, including similar studies that have performed integrated single-cell RNA and ATAC sequencing of multi-stage craniofacial tissues in other vertebrates. There is also concern that some of the conclusions have been over-interpreted, in particular the ability of in silico data to predict the activities of the enhancers tested (since only a minority confirm predictions). Nonetheless, provided these issues can be addressed, this study will represent a significant resource for the craniofacial community.

We thank the reviewer for acknowledging the value of the resources provided by our study, as well as their constructive suggestions, which will further strengthen our manuscript.

Regarding the cell type annotations, the reviewer's reasoning is correct that we retained the terms derived from auto-referencing.

We have now included a number of additional recent references for single-cell studies. If the reviewer had any specific additional reference(s) in mind that should be added, we'd appreciate any additional suggestions.

Please see our point-by-point responses to the reviewer's comments below.

Major Comments

1. Annotation of the mesenchymal cell types in Fig. 3 appears inaccurate. Why is Col2a1 described to label chondrocyte progenitors and not chondrocytes? In Extended Data Fig 3 there is a clear chondrocyte cluster marked by Col2a1, Col9a1, Sox9, etc. More significantly, the cells marked by Msx1 do not appear to be osteoblasts as stated. The cluster marked by Msx1 co-expresses Prrx1, Trps1, Tfp2b which are markers of perichondrium/undifferentiated mesenchyme and not osteoblasts. Better markers of osteoblasts include Ifitm5, Sp7, Ocn but these do not appear in the top 10 list of any cluster. There is also no clear osteoblast cluster in Extended Data Fig 5. One likely explanation is that osteoblasts were not captured in the scRNAseq experiment. Osteoblasts can be difficult to release from mineralized matrix during dissociation and are low or not represented in other single-cell studies of the head.

As correctly recognized by this reviewer in the general comments, by default we retained the labels assigned to the single-cell clusters based on the auto-referencing step. We agree with the reviewer regarding these two clusters and have now replaced the label "chondrocyte progenitors" with "chondrocytes" and the label "osteoblasts" with undifferentiated mesenchyme.

Lastly, what is meant by "skeletal_other"? Teeth, tendon, something else?

We apologize for not including an explanation for this term. The term "skeletal, other" is now replaced with "other craniofacial" as it encompasses a mix of cells with the following descriptive terms from the auto-referencing steps: palate development, roof of mouth, mesenchyme and premature oligodendrocytes. We have also added this note to Methods.

pg 35, lines 785-788, Methods

We note that the label "other craniofacial" encompasses a mix of cells with the following descriptive terms retained from the auto-referencing steps: palate development, roof of mouth, mesenchyme, and premature oligodendrocytes.

General annotations of the mesenchyme clusters lack sophistication and would benefit by comparing to experimental validation annotations in previous single-cell craniofacial studies.

We acknowledge that it has been challenging to further resolve the mesenchymal cluster. In particular, we did not pre-select cells in our study design, for example, by dissecting specific sub-regions of the face or opting for cell populations restricted to a specific lineage. We have now compared the gene expression and accessibility for a subset of annotated marker genes described for the mesenchyme clusters in Fabian et al., 2022 PMID: 35013168 with our own single cell data from the developing mouse face and added details:

pg 24, lines 474-483, Discussion

While previous single-cell studies of the developing face from other animal models have described extensive annotations for ectomesenchyme, we find that the complexity of cell types in the developing mouse face poses some challenges in this respect. In particular, in comparing several mouse orthologs of the embryonic zebrafish ectomesenchymal markers¹⁰¹ expressed in *ScanFaceX* that show relative high accessibility in *ScanFaceN* in neural crest-derived populations (**Extended Data Figure 16**), regional identities marked by specific genes are not obviously delineated in *ScanFaceX*. These differences in cell type distributions and marker gene activity may be explained by the extent of differentiation, growth rate, evolving cell states and developmental timing underlying craniofacial morphogenesis.

2. Line 76 and elsewhere should also cite the many single-cell studies already performed for mouse and zebrafish craniofacial tissues. In particular, Fabian et al, 2022 characterized craniofacial enhancers at multiple embryonic and adult stages with single-cell RNA and ATAC and showed multiple examples of cell- and stage-specific craniofacial enhancers.

Please see response to this reviewer's major comment#1 above.

We have now cited several single-cell studies performed in craniofacial tissues from various model organisms, including Fabian et al., 2022.

pg 4, lines 81-82, pg 5, lines 83-86, Introduction

Several single-cell studies have been performed for the developing face in vertebrate and mammalian model systems, as well as some human face tissues^{9,27,29,31,33,35,37,39,42,45,47,49,51,53,55,57,59,95,101}. While these studies cover several specific cell lineages or anatomical sub-regions of the face, the broad enhancer landscape of mammalian face development at cell type resolution remains incompletely understood.

3. Contrary to what is stated, Extended Data Fig. 9 show little correlation between scRNAseq and snATACseq data for more than half of the clusters, including "osteoblasts".

We agree that our initial description overstated the general extent of correlation. We have now revised the text to the following:

pg 14, lines 296-297, pg 15, lines 298-300, Results

Upon integration, a substantial subset of DARs (10,038 out of 16,564; 60%) across 11 annotated clusters for developing craniofacial cell types were retained. Clusters labeled chondrocytes, myocytes and connective tissue, and sensory neurons showed high correlation between the two data types (**Figure 4a-b, Extended Data Figures 13 and 14; Methods**).

4. Authors should comment on why only 16/60 candidate human enhancers showed craniofacial expression in mouse transgenics, as well as why 13/60 showed expression in other regions but not in the face.

Please see our responses to Reviewer 1, major comment #6 and second part of Reviewer 1, major comment #9 above. In addition, we note that our study did not profile tissues other than craniofacial, and we did not attempt to exclude enhancers from our analyses based on non-craniofacial activities that may have been suggested by other epigenomic datasets (e.g., from ENCODE). Both the tissue-specific validation rate, as well as the occurrence of activity in tissues other than the predicted on (craniofacial) are similar to what we observed in other studies using a combination of ATAC-seq and H3K27ac ChIP-seq for prediction of *in vivo* enhancer activities (see Reviewer 1, major comment #6 for details).

What does it mean that 56% are functionally conserved (from Introduction) since only 16/60 human enhancers had craniofacial activity in mouse transgenic assays?

We apologize for this potentially confusing description. We intended to describe the summary of the overlaps of the 13,893 human candidate enhancers with those from single cell experiments in the mouse. We have now modified this sentence as below, and we hope that it is now clearer for the reader.

pg 3, lines 45-48, Abstract

By integrating these data across species, we find that the majority (56%) of human craniofacial enhancers share single-cell accessible chromatin signatures in the mouse and provide cell type- and embryonic stage-resolved predictions of their *in vivo* activity profiles.

Could there be technical reasons why only 1/4 of human craniofacial enhancers had the expected activity in mouse? For example, it is known that different enhancers have different preferences for minimal promoters.

For a discussion of general success rates of epigenomic enhancer predictions in this assay, please see response to comments from Reviewer1, point #6.

We used the *Shh* enhancer in these transgenic experiments. In our paper introducing the enSERT method for targeted integration of reporter transgenes at the H11 safe harbor locus (Kvon et al., 2020 PMID:32169219), we discuss this choice of promoter and show extensive validation data indicating that this promoter is compatible with a wide range of enhancers and embryonic tissues (see Methods section of Kvon et al.). While we cannot exclude that individual enhancers are not compatible with this promoter, we assume that this is not the predominant reason for the limited success rate, but rather that some enhancers may require a specific broader genomic context (e.g., interaction with other local sequence elements) to show activity and are individually not sufficient to activate transcription when taken out of context to a safe harbor site.

5. Section starting line 140 - how enhancers were linked to nearby genes needs to be better described. Did this leverage combined chromatin/RNA datasets?

Please see response to comment by Reviewer 1, major point#4 above.

6. More data should be provided for the 16 validated human enhancers - how close was their stage- and cell type-specific expression to that predicted by chromatin profiling? In Fig. 5b, it is not convincing that cluster 13 enhancers label chondrocytes. The level of resolution is insufficient to make this claim and many of the patterns look similar to those described for the cluster 2 connective tissue enhancers. For example (but not limited to this example), mm901 appears to label the near entirety of the first and second branchial arches, inconsistent with a specific chondrocyte expression pattern. From the images provided, individual cells cannot be resolved. In Extended Data Fig 11, it is unclear from the resolution provided whether mm2281 labels muscle as predicted. For at least a few cases, co-labeling of mCherry/lacZ with a cell type-specific marker (e.g. Sox9) would help resolve the cell type in which enhancers are active and how this confirms/denies in silico predictions.

We apologize for our unclear description. For the 16 *in vivo* validated enhancers, we tested a single stage e11.5 and analyzed at whole mount resolution. We agree that this resolution is insufficient to decipher stages or exact cell types of activity, and only a broad assessment can be made if the region potentially contains a predicted cell type.

Please see response to the first part of Reviewer1, major comment #9 above.

7. Conclusions from Fig. 6 are over-stated. The scRNAseq analysis of mCherry+ cells only weakly at best corroborates the cell type predictions. The Msx1 and Gbx2 enhancers are most enriched in the "eye, melanocytes" cluster, and Gbx2 mCherry is otherwise very broad in many clusters including "other cellular".

Please see response to second part of comments raised by Reviewer1, point#9 above.

While the mCherry feature plots in Fig. 6a do look close to the RNA feature plots for the corresponding genes in Fig. 6c (strong data), the dot-plots are confusing.

We apologize if the explanations for the dot plots and the conclusions drawn from them were confusing. We have now added a detailed description that walks the reader through the dotplots.

pg 21, lines 401-408, Figure 6, Legend

c. Seurat-based average expression of genes in the vicinity of the respective enhancers, and proportion (percent) of cells expressing those genes in annotated cell types. Enhancer-driven *mCherry* signal is plotted in the center in between the names of the two genes whose promoters are closest to its location within the genome. For example, for hs1431, *mCherry* is highly expressed (indicated by red color intensity) in clusters labeled "other cellular", "myocytes", "skeletal, other", "connective tissue", and "undifferentiated

mesenchyme”, while it is also expressed in a larger proportion of cells (indicated by greater diameter of the circles) in those same clusters. In the same plot, *Snai2* is highly expressed (indicated by blue color intensity) in a subset of cells (indicated by lesser diameter of circles) in identical clusters as compared to *mCherry*.

8. The Discussion is very short and could be expanded to discuss caveats/limitations of the data. For example, Lines 200-201 - can the authors speculate on why 28% of human enhancers sequence conserved in mouse yet mouse enhancers do not show evidence of enhancer activity based on H3K27Ac?

We have now expanded the Discussion to include the caveats/limitations. Please see responses to comments by Reviewer 1, major points #5 and #23, and Discussion.

9. Line 170: how relevant are some of the studies in Supp. Table 6, such as those involved in bitter/sweet/etc taste sensation?

We aimed to be inclusive in our selection of GWAS traits associated with craniofacial development and based our analysis on a keyword search that included terms such as craniofacial, face, cleft lip, cleft palate, microsomia, salivary, tongue, taste, and tooth (see **Methods**). We note that, although rare, disorders of taste sensation (whether of local or systemic etiology) can be tied to developmental origins or disrupted function of craniofacial structures, since tongue development begins in gestational week 4 in humans, continues through the entire embryonic period and involves four pharyngeal arches (Cobourne et al., 2019 PMID: 29784581, Schoenwolf, Gary C., et al. Larsen's human embryology Elsevier Health Sciences, 2020). Enhancers active in the tongue/oral epithelium have been shown to influence morphological traits in mice (Sagai et al., 2017 PMID: 29021530), and we believe it is plausible that enhancers active during embryonic development are causally involved in these traits.

Minor Comments

1. Bottom of graphical abstract, confusing to have UMAPs for cell types and enhancer activity flipped

Thank you for bringing this to our notice. We have now revised the graphical abstract to not mirror the cartoon representation for UMAPs of cell types and enhancer activity (**pg 2**).

2. Fig. 2C,D - unclear where hs2656 is located in tracks.

We apologize for being unclear with this detail. We have now modified Figure 2C to clearly indicate where hs2656 is located in the genomic context shown (**pg 9**).

3. No Author Contribution listed for Bing Ren.

We apologize for missing this detail, author contribution is now listed for Bing Ren.

pg 25, lines 516-517, Author Contributions

B.R. supervised snATAC-seq experiments and integrative analysis of snATAC-seq and scRNA-seq data

Reviewer #3 (Remarks to the Author)

In this study, the authors provide a large dataset of regulatory regions from the developing human face, identifying ~14,000 enhancers from the facial region at embryonic weeks 4 to 8. Using a transgenic mouse reporter assay, they validated numerous putative enhancers that were conserved in facial regions between mouse and human. Using bioinformatics analysis to integrate the human data with single cell RNA-seq and single nuclei-ATAC seq of mouse craniofacial tissue, they speculate that 56% of these enhancers are conserved between mouse and human. Altogether, the work is nicely done and provides a large and useful resource for the community.

While the work seems highly appropriate for Nature Communications, I would recommend modifications as described below.

We thank the reviewer for acknowledging the value of the resources provided by our study, as well as their useful suggestions, which will further enhance our manuscript.

1. The discussion is very short and does not do a particularly good job of putting the present work in context of the existing literature on this topic. In particular, the authors should comment on overlap or lack thereof between the enhancers found in the present study compared with Prescott et al., 2015 and Wilderman et al., 2018.

We have now expanded the Discussion to include the suggested points.

pg 22, lines 441-444, pg 445-458, Discussion

Our compendium of human craniofacial enhancers builds upon previously reported^{26,95} human craniofacial enhancer catalogs, with approximately 5,000 newly identified enhancers for weeks 7-8 of human craniofacial development. When comparing with craniofacial enhancers identified in previous studies, we find that our data provides independent confirmation for 37% of reported primate enhancers and 15% of human-biased enhancers¹⁸. Of the 13,983 reproducible human enhancers described in this study, 47% showed evidence of enhancer-associated RNA signatures in the FANTOM5 database^{96,97}. In contrast, when restricting this analysis to a more differentiated craniofacial cell type available in FANTOM5 (human embryonic palatal mesenchyme)⁹⁶⁻⁹⁸, we observed enhancer RNA signatures were reported for only 3.8% of our 13,983 predicted enhancers, likely reflecting that this cell type is only one of many that were present in our tissue samples (**Supplemental Tables 2 and 21**). Generally, the imperfect overlap of craniofacial enhancers identified in some of these studies may be due to differences in epigenomic profiles from primary tissues comprising the entire face versus *in vitro* differentiation of a specific lineage such as neural crest or palatal mesenchyme.

Additional possible sources of variation include differences in experimental modalities (H3K27ac binding versus measurements of enhancer RNA), and imperfect matching of *in vivo* developmental stages with *in vitro* models.

pg 37, lines 852-860, pg 38 lines 861-863, Methods

Comparing Human Craniofacial Enhancers with Previously Reported Enhancer Catalogs

We compared human enhancers identified in this study with a set of 5,000 primate enhancers profiled from cranial neural crest cell differentiation using both chimpanzee and human cells and a list of 1,000 human-biased enhancers¹⁸. Genomic coordinates of these enhancers were converted to hg38 using LiftOver and intersected with our list of 13,933 reproducible human enhancers. Similarly, enhancers identified by Cap Analysis of Gene Expression (CAGE) including those from normal human embryonic palatal mesenchyme (HEPM: CNhs11894) cells were obtained from the FANTOM5 database⁹⁶⁻⁹⁸. Genomic coordinates of the enhancer lists from FANTOM5 were converted to hg38 and intersected with the 13,983 human reproducible craniofacial enhancers from this study. Results of these analysis are reported in **Supplemental Table 2**.

In addition, please see responses to comments by Reviewer 1, major points# 11c and #22, minor points#11, #12 and #19 above, and those by Reviewer 3, point#3 below.

2. Apropos of the above, it would be very useful for the authors to directly compare their 14,000 enhancers to those identified in previous studies and by Vista. How many are novel other than the few described in the present paper? How many were previously described using older technology?

Please see response to Reviewer 1, major comment #22 above, and **Supplemental Table 21**.

3. On lines 381- 383, the authors state “that while many enhancers are functionally conserved between human and mouse, additional human-specific enhancers that show no functional conservation in mice can be identified by profiling human tissues.” They conclude that these are “human specific”. However, this cannot be concluded without functionally testing these enhancers in both human and mouse cells. Putative human enhancers that fail to provide a signal when tested in mouse are not necessarily “human specific”, as this is a negative result. To demonstrate a positive results, i.e. that these enhancers are human specific, the authors must test those that fail to express in mice in a human system. A simple way to do this would be to differentiate human ES cells into cranial neural crest cells (as done by Prescott et al., 2015) and test the ability of enhancers to mediate reporter expression *in vitro*. Only when they are positive in humans and negative in mouse at all stages can one conclude specificity.

We agree that ideally the difference in activity would also be shown in a human system. However, we currently do not have the ability to perform differentiation of human ES cells into cranial neural crest cells as described by Prescott et al. 2015. Furthermore, we caution that this cell-based

approach does not capture the full developmental complexity of *in vivo* development, which involves a large number of distinct subregional and temporal transitional states, e.g., related to morphogenesis. This difference between systems is exemplified by the somewhat limited overlap between our dataset and the enhancers reported in Prescott et al., 2015 (see previous point). This further supports that enhancers identified *in vivo* are not necessarily active in a cell-based paradigm.

With these limitations in mind, we do agree with the general concern regarding predicted (but not validated) human candidate enhancers in terms of claiming human-specific activity. We have now rephrased this statement to focus on what our data demonstrates, i.e., the presence of epigenomic enhancer signatures in human but not mouse tissue:

pg 22, lines 424-428, Discussion

While many predicted enhancers show conserved epigenomic signatures indicating an active enhancer state in both mouse and human, we also observed elements with human-specific enhancer activity signatures, suggesting that the human but not the mouse ortholog is an active *in vivo* enhancer.

pg 22, lines 438-441, Discussion

While human-specific signatures would need to be validated in suitable human tissue- or cell-based assays to conclusively confirm *bona fide* lineage-specific *in vivo* activity, these data suggest that profiling human tissues is an effective way to identify candidate human-specific enhancers.

4. The methods for obtaining human facial tissue are inscrutable and therefore difficult to evaluate. The authors need to specify what tissue was dissected at each stage and how this was done.

We have now added details, including a cartoon depicting the boundaries of the isolated region.

pg 26, lines 532-533, Methods

Whole face region was dissected (excluding eyes) at HDBR (**Extended Data Figure 1**), and...

pg 26, lines 534-536, Methods

Embryos of both sexes were included in the experiments. However, we did not consider embryo sex as a variable in our studies since craniofacial development is expected to show minimal differences at these early stages of development.

REVIEWER COMMENTS

Reviewer #1 (Remarks to the Author):

This revised version is much improved. The authors have satisfactorily addressed most of my previous comments. This study represents a useful catalogue of datasets and resource for the field. However, in my opinion, it does not provide sufficient new insights into the molecular regulation of craniofacial development to warrant publication in Nature Communications.

Reviewer #2 (Remarks to the Author):

This revision has satisfied most of my concerns except for one important one. I still do not agree that their study has captured "cell-type specific" enhancers as claimed in their title and stated throughout the text. "Cell Type" needs to be removed from the title in the absence of additional experiments. It may also be helpful to add "Human" somewhere in the title as this is the major conceptual and resource advance of the study. In Figure 5b, it remains unclear in which cell types the various enhancer lacZ transgenes are active, as many of the patterns look inconsistent with specific cell types such as chondrocytes. As the authors chose not to perform co-localization of lacZ with known cell type-specific markers, they cannot conclude that these enhancer transgenes label particular cell types rather than general regions of the face. This is also seen in the 3 enhancers tested in Fig. 6. In particular, hs1431 appears very broad in both the lacZ pattern and mCherry clusters - certainly not "cell type-specific" as claimed. I see similar criticisms of this claim of "cell type specificity" in the comments of reviewer 1. The title and text need to be revised to more accurately reflect that this study shows tissue but not cell type specificity in the validation of the bioinformatics analysis.

Reviewer #3 (Remarks to the Author):

The authors have done a good job of revising this manuscript to address most of my concerns. However, I was not satisfied with one minor point raised in my previous review. The authors refer to some enhancers as "human-specific enhancers" without testing their ability to drive expression in human tissue. These regions may be open but repressed so one cannot refer to them as human specific enhancers. They should change this to human specific regions of open chromatin, or something to that effect. I also disagree that this couldn't be tested in an in vitro system. While this may not provide spatiotemporal information, it would at least tell you whether these open chromatin regions have the ability to drive reporter expression. But I would be satisfied with simply correcting the wording.

With these changes, the data provide a useful resource of putative enhancers involved in craniofacial development across species and would be appropriate for publication in Nature Communications.

Point-by-Point Response to Reviewers' Comments

NCOMMS-23-28972A: Cell Type- and Tissue-specific Enhancers in Craniofacial Development by Rajderkar et al.,

We thank the reviewers for their comments on our revised manuscript. We have now made additional changes to the manuscript to address all remaining comments and concerns as described in detail below.

Reviewer #1 (Remarks to the Author)

This revised version is much improved. The authors have satisfactorily addressed most of my previous comments. This study represents a useful catalogue of datasets and resource for the field. However, in my opinion, it does not provide sufficient new insights into the molecular regulation of craniofacial development to warrant publication in Nature Communications.

We thank the reviewer for conveying satisfaction regarding their comments that we addressed in the previous revision.

Reviewer #2 (Remarks to the Author)

This revision has satisfied most of my concerns except for one important one. I still do not agree that their study has captured "cell-type specific" enhancers as claimed in their title and stated throughout the text. "Cell Type" needs to be removed from the title in the absence of additional experiments. It may also be helpful to add "Human" somewhere in the title as this is the major conceptual and resource advance of the study.

We thank the reviewer for their helpful comments. We have now revised the title to remove any references to "Cell Type" and we added "Human" to emphasize one of the major conceptual and resource advances of the study. The new title reads as follows:

Dynamic Enhancer Landscapes in Human Craniofacial Development

In Figure 5b, it remains unclear in which cell types the various enhancer lacZ transgenes are active, as many of the patterns look inconsistent with specific cell types such as chondrocytes. As the authors chose not to perform co-localization of lacZ with known cell type-specific markers, they cannot conclude that these enhancer transgenes label particular cell types rather than general regions of the face. This is also seen in the 3 enhancers tested in Fig. 6. In particular, hs1431 appears very broad in both the lacZ pattern and mCherry clusters - certainly not "cell type-specific" as claimed. I see similar criticisms of this claim of "cell type specificity" in the comments of reviewer 1. The title and text need to be revised to more accurately reflect that this study shows tissue but not cell type specificity in the validation of the bioinformatics analysis.

We have now revised all sections of the manuscript to remove any claims of definitive evidence for "cell type specificity" of enhancer predictions and/or reporter activity as suggested. Specific sections where such changes were made are indicated in blue throughout the manuscript, and are listed below:

pg 2, Graphical Abstract

Caption for the cartoon in lower right corner was changed from "Cell type-specific enhancer activity" to "Single-cell analysis of craniofacial enhancer activity"

pg 3, Abstract, line 30

*Changed to “and **cell type-resolved** in vivo activities”.* – Note that this instance refers to a current knowledge gap, not a statement of what was accomplished in this study.

pg 3, Abstract, lines 46-47

*...and we provide **cell population-** and embryonic stage-resolved predictions.*

pg 5, Introduction, lines 96-97

*...and open chromatin signatures **at single cell resolution** for the developing mouse face.*

pg 12, lines 255-256, Results

*...with the goal to identify the **cell population-resolved activity signatures** of individual enhancers.*

pg 14, line 289, Results, Section Heading

Differential Chromatin Accessibility Correlates with Gene Expression Signatures

pg 17, lines 337-338, Results, Figure 4 Legend Title

Differential chromatin accessibility at craniofacial in vivo enhancers correlates with expression of nearby genes

pg 18, line 353, Results, Section Heading

Cell Population-resolved Enhancer Predictions and in vivo Enhancer Activity Patterns

pg 18, line 360-361, Results

*...the **predicted connective tissue-mesenchymal cluster** (cluster 2)..*

pg 18, lines 368-369, Results

*...which relies on **intricate cellular processes** in combination with highly regionalized regulatory cues.*

pg 19, lines 372-374, Results, Figure 5 Legend

Correlating Cell Population-Resolved Enhancer Signatures with Enhancer in vivo Activity Patterns. a. Heatmap indicates the chromatin accessibility of 77 craniofacial in vivo VISTA enhancers in 11 major clusters representing predicted cell types.

pg 19, line 378, Results, Section Heading

Craniofacial Enhancer Activity at Single-cell Resolution

pg 19, lines 379-380, Results

*To explore whether craniofacial enhancer activity can be quantitatively assigned to **specific cell types in vivo**, - Note that this refers to the general question we are exploring in this section, not the results we obtained.*

pg 19, line 386, Results

*...almost all **cell clusters**,*

pg 20, lines 388-392, Results

*In contrast, **hs746** which is in the vicinity of **Msx1**, is **primarily active in a cluster predicted to represent undifferentiated mesenchyme** and in a subset of cells expressing **Msx1 in ScanFaceX**, a gene previously shown to regulate the osteogenic lineage⁹⁹. **Similarly, based on ScanFaceX***

annotations, enhancer hs521, located near Gbx2, is primarily active in a subset of predicted mesenchymal cells and chondrocytes,

pg 21, lines 400, Figure 6, Legend Title

Figure 6. Enhancer activity at single-cell resolution.

pg 22, line 428, Discussion

including temporal profiles and predictions of cell type specificity.

Reviewer #3 (Remarks to the Author)

The authors have done a good job of revising this manuscript to address most of my concerns. However, I was not satisfied with one minor point raised in my previous review. The authors refer to some enhancers as "human-specific enhancers" without testing their ability to drive expression in human tissue. These regions may be open but repressed so one cannot refer to them as human specific enhancers. They should change this to human specific regions of open chromatin, or something to that effect. I also disagree that this couldn't be tested in an in vitro system. While this may not provide spatiotemporal information, it would at least tell you whether these open chromatin regions have the ability to drive reporter expression. But I would be satisfied with simply correcting the wording.

With these changes, the data provide a useful resource of putative enhancers involved in craniofacial development across species and would be appropriate for publication in Nature Communications.

We thank the reviewer for this suggestion. We have now clarified the wording to address this comment:

pg 11, lines 230-232, Results

*The remaining 3,922 (28%) regions were sequence-conserved but showed no evidence of enhancer activity in the mouse tissues examined (**Supplemental Table 12; Methods**), suggesting that *these regions are active enhancers in humans only.**

pg 11, lines 237-238, Results

...to compare the human and mouse orthologs of an enhancer showing an active enhancer signature in the human genome only.

pg 22, lines 433-434, Discussion

We also provide additional predictions of regions with human-specific enhancer signatures that show no functional conservation in mice that can be identified by profiling human tissues.

pg 22, lines 445-447, Discussion

these data suggest that profiling human tissues is an effective way to identify candidate regions with human-biased enhancer signatures.

REVIEWERS' COMMENTS

Reviewer #2 (Remarks to the Author):

This revised manuscript addresses my remaining concerns and I think the title is now appropriate. This study of human craniofacial enhancers represents a rich resource for the community and in my view is of sufficient advance for publication in Nature Communications.

Reviewer #3 (Remarks to the Author):

I am satisfied with the revisions made by the authors and think the work is now acceptable for publication.